# Improving Semantic Proximity in Information Retrieval through Cross-Lingual Alignment

**Seongtae Hong,  Youngjoon Jang,  Jungseob Lee,  Hyeonseok Moon**[*]**,  Heuiseok Lim**[*]
Department of Computer Science and Engineering, Korea University
{ghdchlwls123, dew1701, omanma1928, glee889, limhseok}@korea.ac.kr

## Abstract

With the increasing accessibility and utilization of multilingual documents, Cross-Lingual Information Retrieval (CLIR) has emerged as an important research area. Conventionally, CLIR tasks have been conducted under settings where the language of documents differs from that of queries, and typically, the documents are composed in a single coherent language. In this paper, we highlight that in such a setting, the cross-lingual alignment capability may not be evaluated adequately. Specifically, we observe that, in a document pool where English documents coexist with another language, most multilingual retrievers tend to prioritize unrelated English documents over the related document written in the same language as the query. To rigorously analyze and quantify this phenomenon, we introduce various scenarios and metrics designed to evaluate the cross-lingual alignment performance of multilingual retrieval models. Furthermore, to improve cross-lingual performance under these challenging conditions, we propose a novel training strategy aimed at enhancing cross-lingual alignment. Using only a small dataset consisting of 2.8k samples, our method significantly improves the cross-lingual retrieval performance while simultaneously mitigating the English inclination problem. Extensive analyses demonstrate that the proposed method substantially enhances the cross-lingual alignment capabilities of most multilingual embedding models.

## 1 Introduction

Information Retrieval (IR) is a core technology that accurately identifies and provides relevant information from large-scale document collections based on user queries. With the recent increase in multilingual data and the growing demand for accessing it, the importance of Cross-Lingual Information Retrieval (CLIR) has become more pronounced (Adeyemi et al., 2024; Guo et al., 2024; Zhang et al., 2023a; Mayfield et al., 2023). Commonly adopted CLIR evaluation settings primarily focus on scenarios where the query language differs from the language of the document pool, which is composed of a single language (Lawrie et al., 2023a; Braschler, 2003). The objective is to measure retrieval performance on a set of documents in a language different from that of query, thereby assessing the cross-lingual representation capability when query and document languages are not equal. Similarly, the Multilingual Information Retrieval (MLIR) task involves retrieving and ranking relevant documents from an integrated collection containing three or more languages (Lawrie et al., 2024).

Despite these efforts, we observe that the practical effectiveness of multilingual embedding models in these settings has not been adequately assessed, and significant performance blind spots remain. Our research question is whether multilingual embedding models can consistently maintain retrieval performance for queries given in different languages, in realistic scenarios where documents in two languages coexist, without performance degradation stemming from cross-lingual semantic misalignment or language-specific biases. In fact, our exploratory analysis reveals that when retrieving from a document pool containing both English and another language that matches the query language, many multilingual embedding models exhibit significant cross-lingual semantic misalignment and language bias toward certain languages, severely degrading retrieval performance (Wu & Dredze, 2020; Park & Lee, 2025; Yang et al., 2024; Elmahdy et al., 2024). For instance, when a query is

---
[*]Corresponding author

written in language A, ideally, both the relevant document written in language A and its semantically equivalent counterpart in English should be ranked at the top of the retrieval results. However, due to a prioritization bias toward English—a high-resource language—irrelevant English documents often appear higher in the rankings, while the correct documents written in language A are relegated to lower ranks. Furthermore, we observe a pronounced language-dependent performance disparity, in which retrieval results vary significantly depending on the language of the query. This phenomenon highlights a critical limitation that is difficult to accurately measure or analyze using conventional evaluation settings. Consequently, our findings underscore the necessity of designing a more practical evaluation environment and corresponding metrics, distinct from existing evaluation settings, to rigorously validate balanced cross-lingual semantic alignment.

In this paper, we define a scenario where documents in two languages coexist to comprehensively evaluate cross-lingual alignment capabilities in retrieval, and introduce a new evaluation metric, Max@R. By assessing embedding models within this scenario and using the proposed metric, we provide a detailed analysis of both cross-lingual alignment aspects and retrieval performance, as well as an in-depth examination of whether there exists an inherent preference for a particular language.

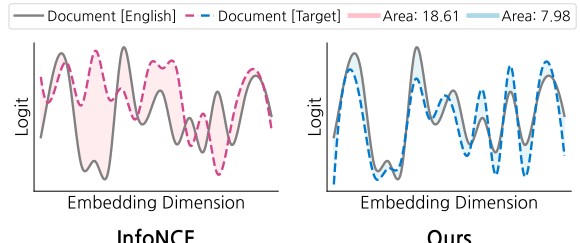

Figure 1: Illustration of distribution-level alignment. Our method shows better alignment than InfoNCE, even when both share the same cosine similarity of 0.99.

Furthermore, based on our analysis of this scenario, we discuss methods to effectively mitigate cross-lingual misalignment. In particular, we propose a unified training strategy that combines two loss functions to jointly optimize cross-lingual alignment and retrieval performance. Specifically, our strategy integrates Jensen-Shannon Divergence (JSD) (Lin, 1991), which aligns the semantic embedding distributions across languages by adjusting the embedding space, with InfoNCE (van den Oord et al., 2019), which directly enhances the retrieval ability between query and document. By applying our proposed method, we demonstrate that even with a relatively small dataset of only 2.8K samples, multilingual embedding models can achieve substantially enhanced cross-lingual alignment and retrieval capabilities. Moreover, we find that performance of our method in monolingual settings, including conventional CLIR, can be maintained or even improved relative to the baseline models.

## 2 PRELIMINARY

In this section, we rigorously define the cross-lingual alignment problem that arises in the evaluation of multilingual embedding models. To accomplish this, we introduce the experimental environment of both CLIR and MLIR, followed by a detailed explanation of the problem definition.

### 2.1 CONVENTIONAL SETTINGS

The objective of a typical IR task is to identify the most relevant documents from a collection in response to a given query. This collection of documents is usually represented as $D = \{d_1, d_2, \ldots, d_n\}$. When a query ($q$) is provided, similarity scores are calculated for the documents within the collection ($D$), and the most similar documents are retrieved based on these scores. CLIR is characterized by the difference between the query language ($L_q$) and the language of the document collection ($L_d$), such that $L_q \neq L_d$. For instance, when a query is provided in a specific language ($L_1$), the goal is to retrieve the most relevant documents written in a different language ($L_2$). MLIR, on the other hand, aims to effectively retrieve the documents from a collection composed of documents in multiple languages ($L_d \in \{L_2, L_3, L_4, \ldots\}$) when the query is presented in a single specific language ($L_q = L_1$).

### 2.2 PROBLEM DEFINITION

Existing evaluation settings primarily address scenarios where the query and document collection languages differ, or where multiple languages are included within documents (Lawrie et al., 2023b). However, these approaches are limited to assessing a model's basic cross-lingual retrieval capabilities

or shallow multilingual retrieval performance, fail to reveal deeper issues such as inaccurate cross-lingual alignment and language bias. Therefore, we consider a multi-reference cross-lingual setup that can comprehensively evaluate both cross-lingual alignment and retrieval performance. For instance, in a scenario where the document pool contains a mixture of two languages, $L_1$ and $L_2$, an effective model should rank all semantically relevant documents at the top, irrespective of the languages used in queries or documents. In other words, documents related to the query, even if expressed in different languages, should be retrieved with equal importance. Ideally, the model should demonstrate equal performance for queries in each language. To simulate an information retrieval scenario in such an environment, we construct and analyze an experimental setup using a dataset that is fully parallel across languages.

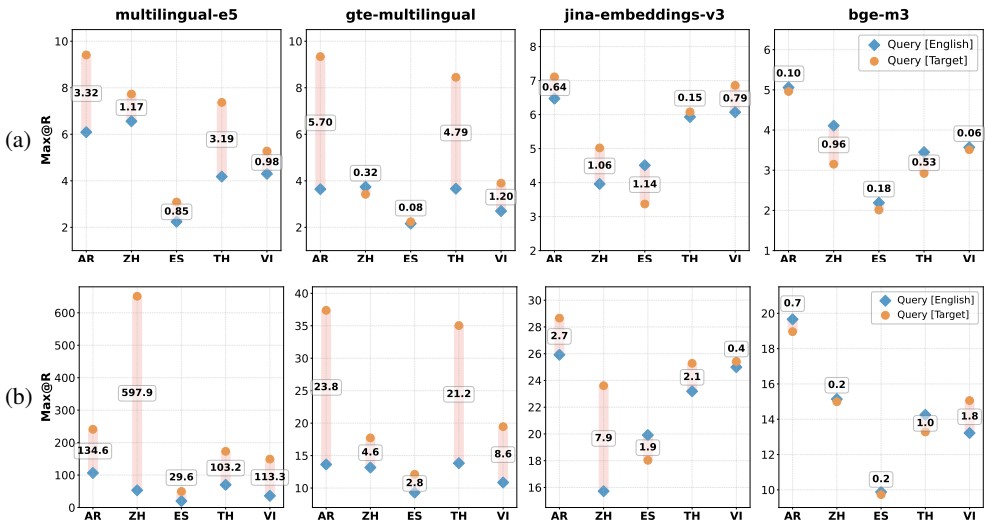

Figure 2: Performance comparison on the XQuAD dataset: (a) CLIR setting, and (b) Multi scenario. We evaluate four multilingual embedding models on five target languages: Arabic (AR), Chinese (ZH), Spanish (ES), Thai (TH), and Vietnamese (VI). Values indicate Max@R gaps.

## 3 MULTI-REFERENCE IN CROSS-LINGUAL INFORMATION RETRIEVAL

Insufficient cross-lingual semantic alignment results in the similarity between a query and documents in other languages not being properly reflected, causing two relevant documents to fail to be ranked at the top. To quantitatively measure this issue and to effectively evaluate performance in this scenario where two languages coexist, we propose a new evaluation metric.

**Max@R**  We evaluate retrieval accuracy for a given query ($q$) using a set of reference documents, denoted as $R_q = \{r_1, r_2, \ldots, r_m\}$. In our multi-reference cross-lingual scenario, these documents are parallel ground-truths written in different languages. For the given query $q$, we denote an ordered list of retrieved documents as $D'(q) = \{d'_1, d'_2, \ldots, d'_n\}$. The documents are sorted by retrieval priority based on semantic similarity to $q$. We define the position $i$ of a document $d'_i$ in the list $D'(q)$ as its rank. To address the limitation that existing metrics (MAP, MRR, or NDCG@k) are not designed to measure when all parallel ground-truths in this multi-reference scenario have been retrieved, we propose Max@R as an essential diagnostic metric. Then, we define **Max@R** as follows:

$$\mathbf{Max@R} = \max(\{i \mid d'_i \in \mathcal{R}_q\}) \tag{1}$$

By calculating Max@R in this manner, Max@R represents the highest (i.e., worst) rank at which all relevant documents in the reference set are first found in retrieval results. This is, in effect, the "actual cutoff point" one must reach to retrieve all documents in $R_q$. In other words, a lower Max@R value indicates that all relevant documents can be retrieved within a smaller portion of the top-ranked results, signifying a more efficient and higher-performing retrieval model.

### 3.1 Limitations of Cross-lingual Information Retrieval in Practical Scenarios

Figure 2 presents the evaluation results of cross-lingual retrieval performance using four different multilingual embedding models, focusing on English as the source language and five target languages. The experiments are conducted under two distinct settings: (a) conventional CLIR, and (b) our proposed multi-reference cross-lingual setup, as defined in Section 2.2. Setting (b) features a document pool with parallel documents in English and the target language, resulting in two ground-truth documents per query. To thoroughly investigate potential biases, we report the performance achieved when querying in both English and the respective target language. Based on these experimental results, we identify and discuss three critical limitations of existing multilingual embedding models and the CLIR environment that were not evident in conventional assessment settings.

**Performance Disparities by Query Language**    One of the most significant limitations observed is the disparity in retrieval performance depending on the query language. In Fig. 2 (a), the overall Max@R is notably low, and the performance differences by query language are relatively minor. In contrast, Fig. 2 (b) shows a substantial gap in performance based on the query language. For the multilingual-e5 model, the results for Chinese (ZH) queries display a marked difference of 597.9 in Max@R compared to the results for English queries. This finding indicates that cross-lingual semantic alignment is inadequate, resulting in amplified performance discrepancies based on the query language in the proposed scenario.

**Representation Instability among Target Languages**    In CLIR, multilingual embedding models tend to exhibit relatively consistent performance across target languages. However, as shown in Fig. 2 (b), there is a significant language-level instability even within the same model. For example, for the gte-multilingual model, the Max@R for Spanish (ES) remains relatively low and stable at 12.12, but rises sharply to 37.38 and 35.04 for Arabic and Thai, respectively, representing nearly a threefold increase. These results indicate that multilingual embedding models exhibit significant semantic representation inconsistency among target languages.

**Excessively High Retrieval Ranks**    In Fig.2 (b), most models and languages exhibit excessively high Max@R values, making them impractical. Notably, for the multilingual-e5 with Chinese (ZH) queries, the Max@R reaches 650.95, meaning that hundreds of documents must be reviewed to find all relevant documents. This result implies that the existing CLIR setting (Fig.2 (a)) fails to capture problems that emerge in practical scenarios, particularly the comprehensive issues of cross-lingual semantic alignment and retrieval capability.

In summary, the proposed scenario and metric have revealed critical issues that were overlooked in conventional settings. This shows that there is still significant room for improvement in multilingual embedding models regarding cross-lingual alignment and retrieval. This underscores the necessity for more realistic and rigorous evaluations across various cross-lingual conditions, thereby providing a clear direction for the advancement of multilingual embedding models.

## 4 Methods

In Section 3.1, we identified the issue of language misalignment within multilingual embedding models. Specifically, the identified problems and corresponding solutions are as follows: First, there is a lack of semantic alignment between text embeddings expressed in different languages, leading to an increased semantic distance between sentences and a decrease in overall retrieval performance. To resolve this cross-lingual semantic misalignment, we align the embedding distributions between English document ($p_{en}$) and its semantically corresponding target language document ($p_{tgt}$) using a loss function based on Jensen–Shannon Divergence (JSD), explicitly optimizing semantic alignment in the embedding space. Second, to mitigate the language bias where retrieval results show an inclination towards documents in a particular language (primarily English), we employ InfoNCE contrastive loss by using positive pairs between English query($q_{en}$) and target language document ($p_{tgt}$). To this end, we utilize a dataset in the form of $(q_{en}, p_{en}, p_{tgt})$ to train the model to minimize a combined loss of (1) JSD-based cross-lingual alignment loss $L_{JSD}$ and (2) InfoNCE contrastive loss $L_{NCE}$, defined as follows:

$$L = \mathbb{E}_{(q_{en}, p_{en}, p_{tgt})}[L_{JSD} + L_{NCE}] \tag{2}$$

## 4.1 Semantic Embedding Alignment via $L_{JSD}$

While standard CLIR objectives are designed to increase the similarity score between a query in one language and a document in another, this approach may be insufficient for achieving robust semantic proximity. Achieving this requires more than just high similarity scores for retrieval; it also necessitates that their underlying embedding representations are fundamentally aligned. As illustrated in Figure 1, representations produced by different objectives can share an identical cosine similarity score yet remain significantly misaligned at the distribution level. To align semantically equivalent passages, our goal is to minimize the divergence between the embedding distributions of texts expressed in two different languages. A prominent method for measuring the disparities between two probability distributions is Kullback-Leibler (KL) divergence (Kullback & Leibler, 1951). It quantifies how a distribution $P$ diverges from another distribution $Q$, and it is defined as follows:

$$D_{KL}(P\|Q) = \sum_i P_i \log \frac{P_i}{Q_i} \tag{3}$$

However, KL-divergence is asymmetric; that is, $D_{KL}(P\|Q) \neq D_{KL}(Q\|P)$. To address this asymmetry, Jensen–Shannon Divergence (JSD) is widely adopted. JSD first defines an intermediate averaged distribution $M$ between two distributions and then measures the average KL-divergence of each distribution from this intermediate distribution. It is defined as follows:

$$\text{JSD}(P\|Q) = \frac{1}{2}D_{KL}(P\|M) + \frac{1}{2}D_{KL}(Q\|M), \quad M = \frac{1}{2}(P+Q) \tag{4}$$

Divergence measures such as KL-divergence and JS-Divergence are widely used to quantify the differences between predicted probability distributions and reference distributions. Previous studies typically focused on modeling the similarities between queries and documents as probability distributions and minimizing the divergence between these predicted distributions and reference distributions. For example, in information retrieval tasks, a model is trained to minimize the distributional divergence between a predicted distribution reflecting query-document similarities and a given reference distribution. In contrast, we propose directly aligning semantic embeddings at the distribution level. Specifically, semantic embedding vectors are interpreted as probability distributions. By explicitly aligning these embedding distributions using JSD, our approach aligns the embedding dimensions across languages, thereby effectively achieving enhanced cross-lingual semantic alignment. To illustrate, let us denote the English document embedding vector as $\mathbf{z}_{d_{en}} \in \mathbb{R}^{\dim}$, and the corresponding target-language embedding vector as $\mathbf{z}_{d_{tgt}} \in \mathbb{R}^{\dim}$, where $\mathbb{R}^{\dim}$ represents the embedding dimension. To transform these embedding vectors into categorical probability distributions over dimensions, we apply the following softmax function:

$$P(\mathbf{z})_i = \frac{\exp(z_i)}{\sum_{k=1}^{\dim} \exp(z_k)}, \quad i = 1, 2\ldots, \dim \tag{5}$$

After transforming the document embedding vectors into probability distributions $P(\mathbf{z}_{d_{en}})$ and $P(\mathbf{z}_{d_{tgt}})$, we perform distribution-level semantic embedding alignment by minimizing the JSD between these probability distributions. The proposed loss function is:

$$\min L_{JSD} = \sqrt{\text{JSD}\big(P(\mathbf{z}_{d_{en}})|P(\mathbf{z}_{d_{tgt}})\big) + \epsilon}. \tag{6}$$

Finally, this loss function employs square root of the Jensen-Shannon divergence, a rigorous distributional distance measure between embedding distributions of two languages. Specifically, taking the square root of JSD satisfies the three distance axioms (identity, symmetry, and triangle inequality), thereby forming a valid metric space (Endres & Schindelin, 2003). Minimizing this distance-based loss during optimization encourages close alignment in the dimension-level probabilistic structures of embeddings, facilitating more effective cross-lingual semantic alignment.

## 4.2 Retrieving Objective via $L_{NCE}$

To improve similarity for cross-lingual query-document pairs, we train the model to maximize the semantic similarity between an English query $q_{en}$ and its corresponding target language passage $p_{tgt}$.

We employ the InfoNCE loss, a representative loss function of contrastive learning, as the objective function. Specifically, the loss for positive pairs $(p_{tgt_i}, q_{en_i}^+)_{i=1}^n$ is defined by the following equation:

$$\min L_{\text{NCE}} = -\frac{1}{n}\sum_i \log \frac{\exp(s(p_{tgt_i}, q_{en_i}^+))}{\exp(s(p_{tgt_i}, q_{en_i}^+)) + \sum_j \exp(s(p_{tgt_i}, q_{en_{ij}}^-))} \tag{7}$$

where $n$ and $m$ denote the total number of data and the batch size, respectively. The negative examples $\{q_{en_{ij}}^-\}_{j=1}^m$ are in-batch negatives (queries of other instances within the same batch). $s(p, q)$ is the relevance score of $p$ and $q$, measured by the cosine similarity between their respective representations. Through the optimization of this contrastive objective function, the semantic similarity between related pairs is maximized, while the similarity with unrelated examples is minimized. This process ultimately facilitates the semantic alignment within an embedding space for cross-lingual retrieval.

## 5 Experiments

In this section, we conduct comprehensive experiments across a variety of scenarios to evaluate the proposed method for cross-lingual alignment capability and retrieval performance.

### 5.1 Experiment Setup

**Scenario**   We design three experimental scenarios: (1) **Multi** evaluates whether the model can retrieve the two ground-truth documents per query in each language from a set of fully parallel documents in two languages. (2) **Multi-1** filters out only the specific same-language ground-truth for each query while retaining the rest of the parallel document pool. This forces the model to identify the corresponding document in the opposite language, evaluating its semantic retrieval capabilities under a strict setting. (3) **Mono** assesses retrieval performance in a single-language environment, further divided into cases where the query and document are in the same language (Mono-Same) and in different languages (Mono-Cross), with the latter corresponding to conventional CLIR. Through these scenarios, we evaluate the reliability of the model in both cross-lingual and monolingual settings.

**Datasets**   To evaluate the three cross-lingual retrieval scenarios, it is essential to use datasets that are fully parallel across languages. This fully parallel structure is a necessary prerequisite for rigorously validating our scenarios and the Max@R metric. This means that the same question-document pairs must exist in multiple languages to enable the evaluation of retrieval performance across different languages. For this purpose, we utilize the multilingual benchmarks XQuAD (Artetxe et al., 2020) and Belebele (Bandarkar et al., 2024).These are high-quality, human-translated benchmarks, widely adopted in standard multilingual retrieval evaluation benchmarks Enevoldsen et al. (2025). A comprehensive description of these datasets and our rationale is provided in Appendix B. Both datasets ensure complete parallelism, making them suitable for comprehensively validating performance across the proposed scenarios.

**Metric**   We employ evaluation metrics that align with the characteristics of each retrieval scenario environment. For the Multi scenario, we utilize Complete@K (Qu et al., 2024), which considers an answer correct only if all relevant documents are included within the top-k results. This metric is reported as a percentage on a 0–100 scale. Additionally, we use Max@R, as proposed in Section 3, and introduce an intuitively interpretable and generalized metric, Max@R$_{\text{norm}}$, which normalizes the varying Max@R values across different datasets on a logarithmic scale. This metric normalizes the maximum rank for each query to a value between 0 and 100, represented as Max@R$_{\text{norm}}$ = $\frac{1}{|\mathcal{Q}|}\sum_{q \in \mathcal{Q}}[100 \times \frac{\log_2(|D|) - \log_2(Max@R)}{\log_2(|D|) - \log_2(|\mathcal{R}|)}]$, where $|D|$ is the size of the document pool, and $|\mathcal{R}|$ is the number of ground-truth documents for each query. For the Multi-1 and Mono scenarios, where there is only one correct document per query, we evaluate retrieval performance using metrics such as NDCG@1 (Järvelin & Kekäläinen, 2002) and MRR (Nogueira & Cho, 2019; Khattab & Zaharia, 2020; Xiong et al., 2020; Karpukhin et al., 2020). By employing appropriate metrics for each scenario, we systematically compare and analyze the proposed method across various scenarios, thereby verifying its validity from multiple perspectives.

Table 1: Main performance results under the **Multi scenario**

| Doc | Query | XQuAD | | | | | | Belebele | | | | | |
|---|---|---|---|---|---|---|---|---|---|---|---|---|---|
| | | Comp@10 | | Max@R ($\downarrow$) | | Max@R$_{norm}$ | | Comp@10 | | Max@R ($\downarrow$) | | Max@R$_{norm}$ | |
| | | Base | Ours | Base | Ours | Base | Ours | Base | Ours | Base | Ours | Base | Ours |
| *multilingual-e5-base* | | | | | | | | | | | | | |
| En+Ar | En | 15.46 | **60.34** | 106.42 | **18.56** | 43.88 | **68.54** | 26.22 | **87.22** | 187.78 | **17.46** | 33.23 | **68.14** |
| | Ar | 8.91 | **53.53** | 241.06 | **30.77** | 32.33 | **61.40** | 15.44 | **75.67** | 231.79 | **29.55** | 30.13 | **60.41** |
| En+Zh | En | 29.24 | **61.60** | 53.04 | **17.35** | 53.71 | **69.50** | 63.44 | **86.78** | 44.74 | **15.96** | 54.31 | **69.47** |
| | Zh | 0.50 | **55.88** | 650.95 | **23.10** | 18.31 | **65.45** | 3.11 | **82.44** | 476.45 | **18.55** | 19.54 | **67.26** |
| En+Es | En | 58.99 | **65.63** | 19.83 | **15.38** | 67.61 | **71.19** | 68.67 | **88.89** | 49.14 | **13.44** | 52.94 | **71.99** |
| | Es | 36.30 | **62.52** | 49.46 | **18.14** | 54.70 | **68.87** | 55.44 | **85.22** | 69.81 | **16.50** | 47.77 | **68.98** |
| En+Th | En | 24.37 | **57.23** | 69.94 | **18.81** | 49.81 | **68.35** | 41.11 | **87.33** | 112.49 | **15.44** | 40.76 | **69.95** |
| | Th | 11.76 | **46.81** | 173.13 | **33.66** | 37.01 | **60.13** | 22.44 | **75.89** | 188.02 | **29.60** | 33.21 | **60.39** |
| En+Vi | En | 45.88 | **63.53** | 35.95 | **17.31** | 59.20 | **69.53** | 56.44 | **88.89** | 69.83 | **12.92** | 47.77 | **72.58** |
| | Vi | 18.15 | **59.50** | 149.21 | **24.15** | 39.11 | **64.82** | 33.44 | **82.89** | 153.05 | **15.54** | 36.23 | **69.86** |
| *gte-multilingual-base* | | | | | | | | | | | | | |
| En+Ar | En | 66.05 | **67.23** | 13.63 | **12.64** | 72.90 | **73.97** | 87.44 | **88.56** | 10.03 | **9.17** | 76.29 | **77.62** |
| | Ar | 49.58 | **52.10** | 37.38 | **31.38** | 58.65 | **61.12** | 79.67 | **80.33** | 26.23 | **23.55** | 62.16 | **63.75** |
| En+Zh | En | 68.15 | **68.99** | 13.16 | **12.51** | 73.40 | **74.11** | 91.00 | **91.56** | 9.23 | **8.90** | 77.52 | **78.06** |
| | Zh | 58.82 | **63.11** | 17.73 | **15.24** | 69.19 | **71.32** | 86.67 | **88.56** | 13.93 | **12.80** | 71.47 | **72.71** |
| En+Es | En | 77.14 | **78.40** | 9.30 | **8.92** | 78.29 | **78.89** | 90.44 | **93.22** | 8.77 | **6.50** | 78.27 | **82.68** |
| | Es | 71.01 | **73.78** | 12.12 | **10.51** | 74.56 | **76.57** | 88.11 | **90.78** | 14.18 | **11.06** | 71.21 | **74.86** |
| En+Th | En | 64.96 | **67.48** | 13.83 | **13.02** | 72.69 | **73.55** | 88.67 | **89.33** | 9.97 | **9.03** | 76.39 | **77.84** |
| | Th | 46.13 | **51.34** | 35.04 | **30.50** | 59.57 | **61.53** | 77.11 | **78.67** | 29.97 | **25.30** | 60.21 | **62.69** |
| En+Vi | En | 73.11 | **73.19** | 10.85 | **10.78** | 76.12 | **76.22** | 90.56 | **91.33** | 8.33 | **7.71** | 79.03 | **80.16** |
| | Vi | 62.10 | **64.71** | 19.46 | **16.75** | 67.87 | **69.99** | 86.33 | **87.56** | 11.63 | **9.94** | 74.12 | **76.43** |
| *jina-embeddings-v3* | | | | | | | | | | | | | |
| En+Ar | En | 55.13 | **70.92** | 25.92 | **11.75** | 63.82 | **74.99** | 85.11 | **90.67** | 19.83 | **9.91** | 66.28 | **76.47** |
| | Ar | 57.65 | **65.04** | 28.66 | **19.29** | 62.41 | **68.00** | 82.78 | **86.22** | 22.93 | **16.47** | 64.14 | **69.10** |
| En+Zh | En | 65.46 | **72.44** | 15.72 | **10.96** | 70.88 | **75.98** | 89.78 | **91.56** | 14.30 | **10.00** | 71.07 | **76.34** |
| | Zh | 58.57 | **70.67** | 23.61 | **13.53** | 65.14 | **73.00** | 85.33 | **91.44** | 18.52 | **11.51** | 67.28 | **74.27** |
| En+Es | En | 68.32 | **75.63** | 19.92 | **10.10** | 67.54 | **77.14** | 85.11 | **92.22** | 24.16 | **9.88** | 63.38 | **76.52** |
| | Es | 68.74 | **73.53** | 18.03 | **11.44** | 68.95 | **75.37** | 88.78 | **90.22** | 18.67 | **12.59** | 67.16 | **72.96** |
| En+Th | En | 56.39 | **71.51** | 23.19 | **11.59** | 65.39 | **75.19** | 86.44 | **91.89** | 17.63 | **9.77** | 68.03 | **76.69** |
| | Th | 58.07 | **68.32** | 25.28 | **16.41** | 64.18 | **70.28** | 84.33 | **86.67** | 22.17 | **15.78** | 64.65 | **69.63** |
| En+Vi | En | 59.75 | **71.43** | 24.99 | **11.69** | 64.34 | **75.07** | 83.00 | **91.89** | 22.91 | **9.91** | 64.16 | **76.48** |
| | Vi | 61.51 | **67.06** | 25.42 | **18.64** | 64.10 | **68.48** | 87.33 | **90.67** | 19.88 | **12.27** | 66.25 | **73.33** |
| *bge-m3* | | | | | | | | | | | | | |
| En+Ar | En | 59.50 | **68.32** | 19.66 | **14.87** | 67.73 | **71.67** | 86.44 | **89.56** | 13.78 | **10.18** | 71.62 | **76.08** |
| | Ar | 65.71 | **67.98** | 18.96 | **17.39** | 68.24 | **69.46** | 85.11 | **85.44** | 18.91 | **15.82** | 66.98 | **69.60** |
| En+Zh | En | 67.82 | **71.09** | 15.14 | **13.08** | 71.42 | **73.48** | 89.44 | **90.56** | 12.14 | **10.90** | 73.49 | **75.07** |
| | Zh | 63.19 | **68.49** | 14.99 | **13.01** | 71.56 | **73.56** | 88.67 | **90.00** | 12.65 | **11.56** | 72.88 | **74.20** |
| En+Es | En | 77.90 | **78.82** | 9.88 | **9.36** | 77.44 | **78.21** | 91.78 | **93.00** | 10.25 | **9.29** | 75.98 | **77.43** |
| | Es | **77.65** | 77.39 | 9.73 | **9.72** | 77.66 | **77.68** | 91.11 | **92.56** | 10.70 | **9.92** | 75.34 | **76.46** |
| En+Th | En | 67.90 | **71.34** | 14.25 | **12.67** | 72.27 | **73.93** | 90.89 | **92.11** | 10.24 | **9.03** | 75.99 | **77.83** |
| | Th | 67.23 | **70.08** | 13.28 | **12.95** | 73.27 | **73.63** | 87.00 | **87.67** | 16.73 | **14.57** | 68.78 | **70.81** |
| En+Vi | En | 71.01 | **73.53** | 13.22 | **11.74** | 73.33 | **75.01** | 90.56 | **91.67** | 10.53 | **8.97** | 75.59 | **77.93** |
| | Vi | 66.72 | **70.92** | 15.06 | **13.40** | 71.49 | **73.14** | 89.00 | **90.78** | 11.01 | **9.80** | 74.93 | **76.63** |

**Implementation Details**   The experiments are conducted on a total of 10 languages. In the main results, we report for five languages: Arabic (AR), Chinese (ZH), Spanish (ES), Thai (TH), and Vietnamese (VI). Detailed experiments for the remaining five languages, German (DE), Greek (EL), Hindi (HI), Romanian (RO), and Turkish (TU), are provided in the Appendix D. We employ four multilingual embedding models for the experiments: multilingual-E5-base (Wang et al., 2024), gte-Multilingual-base (Zhang et al., 2024), jina-embeddings-v3 (Sturua et al., 2024), and bge-M3 (Chen et al., 2024). We utilize the MIRACL train dataset (Zhang et al., 2023b), which consists of 2.8k English query-document pairs. To obtain documents in each target language, the positive documents in English are translated into each target language using the GPT-4o (OpenAI, 2024). Additional details regarding the training procedure are provided in Appendix C.

## 5.2 MAIN RESULTS

In this section, we evaluate our method within a multi-scenario, where documents in English and the target language coexist, and each query has two gold documents. The results, presented in Table 1, are analyzed by querying in both English and the target language.

**Enhanced Cross-lingual Alignment**   The proposed method demonstrates clear and consistent improvements over baseline models across all evaluated language pairs and metrics. Notably, based on Complete@10, while baseline models exhibit relatively limited performance when queried in non-English languages, our proposed method achieves significant performance enhancements for all languages and models considered. These improvements indicate that our method effectively facilitates semantic alignment among different languages within an embedding space. Ultimately, these results confirm that our method successfully addresses the limitations of existing multilingual embedding models, which are not captured by the conventional CLIR setting.

**Reducing Variance in Language Bias**   In addition to improving retrieval performance, the proposed method reduces the issue of performance disparity between languages. Baseline models typically exhibit a bias toward English data, resulting in significantly reduced performance for non-English queries. However, the proposed method results in a relative decrease in the quantitative performance disparity between English and target languages. For instance, the language performance gap for jina-embeddings-v3 (En+Zh) consistently decreases from $6.89\%p \rightarrow 1.77\%p$ on XQuAD and from $4.45\%p \rightarrow 0.12\%p$ on Belebele. This result demonstrates that the model has mitigated the previous misalignment between English and target languages, reducing language bias and contributing to improved language equity.

**Enhanced Full-Recall Ranking Performance**   To intuitively assess the cross-lingual alignment, we focus on analyzing the Max@R metric in our experiments. The results show that our method consistently achieves significantly lower Max@R scores across all languages and datasets compared to the baseline. For instance, with Chinese queries on the multilingual-e5-base model, Max@R significantly improves from 650.95 to 23.10 on XQuAD. This indicates that our method is effective in positioning relevant passages near the top, demonstrating robust performance in cross-lingual multi-reference scenarios. These improvements are also observed by consistent gains on the Max@R$_{norm}$.

## 5.3   CASE STUDY

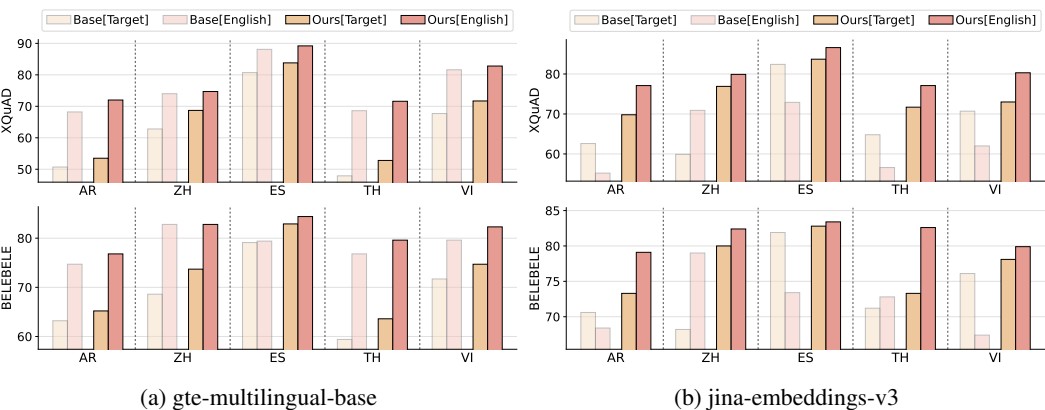

(a) gte-multilingual-base                    (b) jina-embeddings-v3

Figure 3: NDCG@1 comparison in the **Multi-1 scenario** with [lang] as the query language

**Cross-Lingual Information Retrieval in the Multi-1 Scenario**   The Multi-1 scenario provides a more rigorous assessment of cross-lingual semantic alignment capabilities. As shown in Figure 3, existing baselines encounter substantial difficulties in retrieving a relevant document written in languages different from the query language. In contrast, our proposed method consistently improves the NDCG@1 across all language pairs, demonstrating enhanced cross-lingual semantic proximity within the semantic embedding space. Furthermore, the observed performance gains are consistent and substantial, irrespective of whether the queries are in English or in the respective target languages.

**Performance Validation in Monolingual Settings**   To confirm whether the proposed method maintains robust retrieval performance in monolingual environments, we conduct evaluations under two settings: Mono-Same and Mono-Cross. The results, summarized in Table 2, demonstrate that our approach largely preserves or even modestly improves upon the baseline models in the Mono-Same

Table 2: **Mono scenario** performance using jina-embeddings-v3: English / Target language queries.

| Lang | Model | XQuAD | | | | Belebele | | | |
|---|---|---|---|---|---|---|---|---|---|
| | | Mono-Same | | Mono-Cross | | Mono-Same | | Mono-Cross | |
| | | NDCG@1 | MRR | NDCG@1 | MRR | NDCG@1 | MRR | NDCG@1 | MRR |
| Ar | Base | 89.2/81.9 | 0.933/0.879 | 79.5/80.6 | 0.863/0.874 | 88.4/83.0 | 0.919/0.873 | 80.1/77.0 | 0.856/0.833 |
| | Ours | **90.8/83.9** | **0.945/0.898** | **87.1/81.9** | **0.919/0.884** | **88.4/87.0** | **0.922/0.871** | **82.6/79.8** | **0.880/0.856** |
| Zh | Base | 89.2/88.2 | 0.933/0.926 | 83.3/83.8 | 0.895/0.895 | **88.4**/86.9 | 0.919/0.912 | 83.4/81.9 | 0.882/0.874 |
| | Ours | **90.7/89.5** | **0.943/0.937** | **88.0/86.5** | **0.927/0.916** | 88.1/**87.0** | **0.920/0.912** | **84.9/84.4** | **0.895/0.896** |
| Es | Base | 89.2/85.6 | 0.933/0.907 | 85.3/87.5 | 0.905/0.922 | **88.4**/84.9 | **0.919**/0.895 | 83.4/84.4 | 0.882/0.890 |
| | Ours | 89.2/**88.8** | **0.936/0.931** | **89.3/88.0** | **0.935/0.925** | 87.7/**85.4** | 0.918/**0.898** | **85.6/84.7** | **0.901/0.892** |
| Th | Base | 89.2/82.4 | 0.933/0.885 | 80.4/81.7 | 0.873/0.879 | 88.4/82.2 | 0.919/0.875 | 80.7/78.6 | 0.865/0.844 |
| | Ours | **91.2/86.4** | **0.947/0.916** | **87.6/83.1** | **0.925/0.894** | **88.7/82.2** | **0.923/0.876** | **85.2/78.6** | **0.899/0.848** |
| Vi | Base | **89.2**/84.4 | **0.933**/0.897 | 79.4/81.1 | 0.862/0.878 | **88.4**/86.1 | **0.919/0.900** | 80.7/82.1 | 0.859/0.873 |
| | Ours | 88.9/**86.9** | 0.933/**0.918** | **87.6/81.8** | **0.922/0.884** | 87.6/85.0 | 0.914/0.897 | **83.4/83.6** | **0.889/0.887** |

scenario. Notably, slight performance gains are observed for target language queries, suggesting that the quality of single language semantic representations is indirectly enhanced via our alignment method. In the Mono-Cross scenario, our method further surpasses the baseline models, providing consistent improvements. Collectively, these findings indicate that our approach does not compromise and sometimes even improves, monolingual retrieval performance. This affirms that reducing language misalignment in embedding spaces improves representational quality and retrieval performance in both cross-lingual and monolingual settings.

## 5.4 ABLATION STUDY

In this section, we perform an ablation study to clearly examine the roles of two loss components in our method: the Jensen–Shannon divergence-based embedding alignment loss $L_{JSD}$ and the InfoNCE-based query-document relevance learning loss $L_{NCE}$. We also include a comparative experiment, $L_{NCE_{psg}}$, on enhancing the similarity between English documents $(p_{en})$ and target language documents $(p_{tgt})$ (Feng et al., 2020; Chi et al., 2020).

Table 3: Ablation results for loss components in **Multi scenario** experiments on the Belebele dataset. The results are reported in Max@R$_{norm}$ for both English and target language queries.

| Methods | Ar | Zh | Es | Th | Vi |
|---|---|---|---|---|---|
| | | | gte-multilingual-base | | |
| Baseline | 76.29 / 62.16 | 77.52 / 71.47 | 78.27 / 71.21 | 76.39 / 60.21 | 79.03 / 74.12 |
| $L_{NCE_{psg}}$ | 75.45 / 63.28 | 76.07 / 71.72 | 79.33 / 74.13 | 76.34 / 62.53 | 77.79 / 74.77 |
| Ours | **77.62 / 63.75** | **78.06 / 72.71** | **82.68 / 74.86** | **77.84 / 62.69** | 80.16 / **76.43** |
| w/o $L_{JSD}$ | 75.63 / 63.19 | 75.18 / 71.42 | 80.43 / 74.14 | 74.75 / 61.93 | 78.99 / 75.39 |
| w/o $L_{NCE}$ | 76.14 / 59.96 | 78.01 / 70.09 | 82.00 / 71.66 | 77.13 / 59.65 | **80.20** / 73.84 |
| | | | jina-embeddings-v3 | | |
| Baseline | 66.28 / 64.14 | 71.07 / 67.28 | 63.38 / 67.16 | 68.03 / 64.65 | 64.16 / 66.25 |
| $L_{NCE_{psg}}$ | 72.37 / 68.12 | 75.00 / 72.92 | 68.06 / 69.93 | 72.35 / 68.66 | 70.52 / 70.46 |
| Ours | **76.47 / 69.10** | **76.34 / 74.27** | **76.52 / 72.96** | **76.69 / 69.63** | **76.48 / 73.33** |
| w/o $L_{JSD}$ | 71.58 / 67.73 | 74.64 / 72.96 | 68.31 / 69.88 | 71.90 / 68.19 | 70.42 / 70.19 |
| w/o $L_{NCE}$ | 40.68 / 34.29 | 37.52 / 38.29 | 57.44 / 54.18 | 15.47 / 14.26 | 43.73 / 42.41 |

Results in Table 3 indicate clearly complementary roles for these components. Specifically, the absence of $L_{JSD}$ negatively affects cross-lingual semantic embedding alignment and overall retrieval performance, whereas excluding $L_{NCE}$ limits the retrieval effectiveness, despite embedding-level alignment. This demonstrates that the combination of $L_{JSD}$ and $L_{NCE}$ is essential for effectively achieving objectives of semantic alignment within an embedding space and securing retrieval performance. It also proves that relying solely on one of these components is insufficient to enhance overall cross-lingual retrieval.

Additionally, we analyze $L_{NCE_{psg}}$. The results show that this approach generally improves retrieval performance over the Baseline. confirming that enhancing document-level similarity is a valid strategy for improving cross-lingual representations. Crucially, regardless of the base model, Ours consistently and significantly outperforms the $L_{NCE_{psg}}$ approach. This result highlights the source of our method's superiority: rather than simply increasing a similarity score between documents, it provides a more fundamental solution by directly aligning the distribution of the output representations themselves, making it more effective for the end task of query-document retrieval.

## 6 RELATED WORKS

Most existing studies in Cross-Lingual Information Retrieval (CLIR) focus on bridging the semantic gap by constructing cross-lingual embedding spaces or leveraging knowledge transfer approaches to minimize query-document distances across languages (Huang et al., 2023a; Yu et al., 2021; Litschko et al., 2021; Valentini et al., 2025; Lin et al., 2023). Unsupervised methods, in particular, have attempted to reduce dependencies on translation resources by training shared embedding spaces only from monolingual corpora (Litschko et al., 2018). Concurrently, several studies address low-resource languages, where parallel corpora are limited, by proposing optimal transport-based knowledge distillation or multi-stage knowledge distillation techniques to transfer ranking knowledge from high-resource languages (Huang et al., 2023a;b). Additionally, integrating knowledge graphs into query-document representations has shown promise in alleviating cross-lingual semantic gaps (Zhang et al., 2022; Litschko et al., 2022). Collectively, these studies underscore that aligning and refining cross-lingual embedding representations is critically important for CLIR. However, most prior studies assume either purely monolingual or entirely multilingual document pool, thus failing to adequately address biases and misalignments that arise when two languages coexist in a single document pool.

In parallel, studies have increasingly focused on explicit approaches for aligning multilingual embedding spaces. Hu et al. (2020) leverages parallel corpora and explicit alignment objectives to enhance sentence-level cross-lingual transferability. More granular studies addressing contextual embedding alignment have introduced nuanced evaluation tasks, such as dependency parsing or token-level semantic retrieval, highlighting the importance of fine-grained measures (Schuster et al., 2019; Liu et al., 2019). While insightful, these works primarily emphasize alignment in general representation tasks, offering limited consideration of practical retrieval challenges associated with combined-language environments.

## 7 CONCLUSION

To investigate severe semantic misalignment and language disparities exhibited by existing multilingual embedding models, we propose a new evaluation scenario and a metric, Max@R. Through our experiments, we reveal previously unseen issues that were not observable using existing evaluation scenarios. To address these issues, we present a training strategy that effectively achieves semantic proximity in the cross-lingual embedding space by leveraging Jensen-Shannon Divergence for semantic embedding alignment and InfoNCE for enhancing cross-lingual retrieval performance. Our method mitigates linguistic misalignment and language bias, significantly improving cross-lingual retrieval performance and effectively reducing performance disparities across languages. Additionally, our method demonstrates stable performance even in monolingual settings.

## ETHICS STATEMENT

This study proposes a cross-lingual embedding alignment methodology that can positively impact applications requiring effective information retrieval across multiple languages. The proposed method accurately aligns semantic representations across different languages to enhance retrieval performance, while simultaneously strengthening the generalization and robustness of multilingual embedding models. In particular, our approach is applicable even to low-resource languages, potentially mitigating global information disparity and contributing to the creation of a more equitable information access environment. Moreover, by effectively utilizing existing large-scale embedding models and data resources, our approach significantly reduces additional costs associated with data construction and labeling. However, because the training process involves translations generated by large language

models, there are potential risks of subtle cultural nuances being distorted or the introduction of data biases, which may lead to inaccurate or unintended outcomes for certain cultural or linguistic groups. This study transparently acknowledges and carefully discusses these limitations and risks. Nevertheless, we firmly believe that the expected benefits and positive impacts of our research substantially outweigh the aforementioned concerns.

## REPRODUCIBILITY STATEMENT

Our research is designed for full reproducibility. Details on the experimental setup, including datasets and implementation specifics, are described in Section 5.1. Further training information, such as the specific hyperparameters used, can be found in Appendix C. For the computational resources required to run our experiments, please refer to Appendix C.

## ACKNOWLEDGMENTS

This research was supported by Basic Science Research Program through the National Research Foundation of Korea(NRF) funded by the Ministry of Education(NRF-2021R1A6A1A03045425). This work was supported by Institute for Information & communications Technology Promotion(IITP) grant funded by the Korea government(MSIT) (RS-2024-00398115, Research on the reliability and coherence of outcomes produced by Generative AI). This work was supported by Institute of Information & communications Technology Planning & Evaluation (IITP) under the artificial intelligence star fellowship support program to nurture the best talents (IITP-2026-RS-2025-02304828) grant funded by the Korea government(MSIT). This work was supported by ICT Creative Consilience Program through the Institute of Information & Communications Technology Planning & Evaluation(IITP) grant funded by the Korea government(MSIT) (IITP-2026-RS-2020-II201819)

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

## A  LIMITATIONS

A major limitation of this study arises from the experimental setup and evaluation process for cross-lingual alignment, which are primarily centered around English despite the extensive possibilities of language combinations. In real-world scenarios, various language pairs that do not involve English frequently exist, requiring consideration. Nonetheless, experiments and evaluations in this study were intentionally centered around English, the most widely used and high-resource language, under the assumption that multilingual models would exhibit significant bias toward alignment with English, and to reflect the most practical real-world scenarios.

Additionally, although our evaluations considered various scenarios (Multi, Multi-1, Mono) to assess cross-lingual retrieval performance, the current experimental design focused mainly on two languages settings. In practice, multilingual contexts involving more than two languages frequently occur. However, we restricted our evaluations primarily to cross-lingual settings because existing models still struggle with performance even in these simpler setups.

Moreover, our approach utilized machine translation based on language models to build the training dataset. Compared to human translation, automatically translated data may fail to capture subtle linguistic nuances and cultural contexts sufficiently Toral & Way (2018); Läubli et al. (2018); Lee et al. (2024). However, we adopted this methodology as it currently represents the most practical and efficient solution available for constructing large-scale multilingual datasets.

## B  EVALUATION BENCHMARK DETAILS

In this study, we utilize multilingual Question-Answering (QA) datasets with parallel structures, converted into retrieval tasks, to evaluate cross-lingual retrieval performance. Specifically, we use XQuAD[1], a multilingual question-answer dataset derived from SQuAD 1.1 Rajpurkar et al. (2016), which consists of fully parallel question-answer pairs across 13 languages including English. XQuAD was directly translated by professional translators, ensuring a precise one-to-one correspondence between documents and queries across languages. This high-quality translation process naturally preserves linguistic expressions and semantic meanings in each target language, making XQuAD particularly suitable for assessing the robustness of embedding models against language variations in cross-lingual scenarios. Additionally, we utilize Belebele[2], another multilingual QA dataset that includes diverse language pairs. Belebele was carefully translated from English into multiple languages by native-speaking translators proficient in English, effectively capturing contextual nuances and cultural subtleties Bandarkar et al. (2024). Due to these characteristics, Belebele provides realistic and varied scenarios reflecting practical multilingual retrieval environments, enabling fine-grained comparison and analysis of retrieval performance across languages. Furthermore, our use of these specific benchmarks aligns with standard practices for robust multilingual evaluation. Both XQuAD and Belebele are core benchmarks used for assessing multilingual retrieval capabilities in the prominent Massive Multilingual Text Embedding Benchmark (MMTEB) Enevoldsen et al. (2025)[3].

## C  TRAINING DETAILS

**Dataset Translation**   We employed a Large Language Model to translate datasets for training purposes. The format template for translation prompts used as inputs to the model is as follows:

```
System: #Instructions
        Translate the following English passage fully and
        accurately into {target language}

User: {English passage}
Assistant: {Translated passage}
```

---

[1]https://huggingface.co/datasets/google/xquad
[2]https://huggingface.co/datasets/facebook/belebele
[3]https://github.com/embeddings-benchmark/mteb

**Hyperparams**  All models were trained for a total of one epoch with a batch size of 32, employing a linear learning rate scheduler with a warm-up ratio of 0.15. We used the AdamW optimizer (Loshchilov & Hutter, 2019) (with parameters $\beta_1 = 0.9$, $\beta_2 = 0.99$, and weight decay=0.01), and adopted bfloat16 mixed precision for computational efficiency. Considering the characteristics of each model, we set the initial learning rates as follows: 4e-6 for bge-M3, 2e-5 for multilingual-E5-large, 1e-5 for gte-multilingual-base, and 3e-5 for jina-embeddings-v3.

**Hardware**  We used 2 NVIDIA A100 GPUs, each with 80GB of memory capacity, along with AMD EPYC 7513 processors featuring 32 cores, to train models. For evaluation, we employed a single accelerator.

**Reproducibility**  To ensure a fair and consistent comparison, all experimental results reported in this paper are based on a single run using a fixed random seed (42). This fixed seed controls all stochastic elements, including data shuffling and the batch composition and sampling order. This approach allows us to strictly control the training environment for both baselines and our proposed model, ensuring that the comparison fairly isolates the effect of our methodology.

# D  EXTENDED RESULTS FOR ADDITIONAL LANGUAGES

In this section, we provide experimental results for three scenarios involving five languages not previously included in the main body of the paper: German (DE), Greek (EL), Hindi (HI), Romanian (RO), and Turkish (TU).

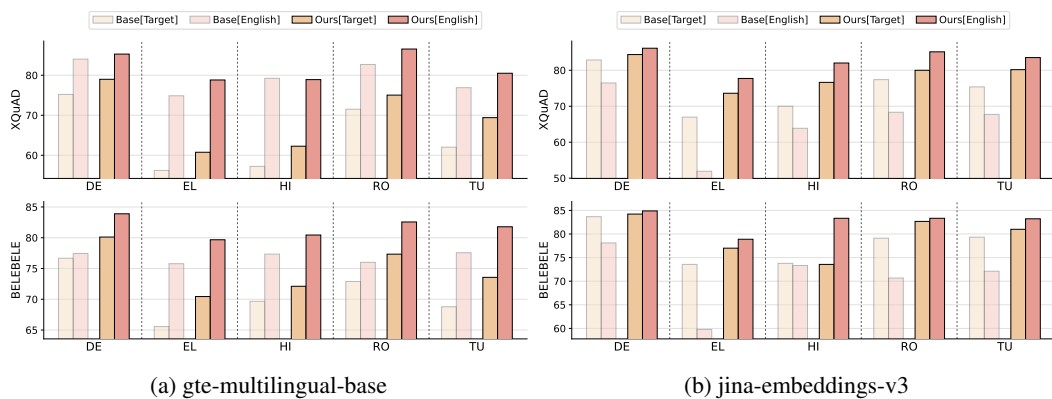

(a) gte-multilingual-base  (b) jina-embeddings-v3

Figure 4: NDCG@1 comparison on additional language pairs in the **Multi-1 scenario** with [lang] as the query language.

Table 4: Performance comparison on remaining language pairs under the **Multi scenario**.

| Doc | Query | XQuAD Comp@10 Base | Ours | Max@R (↓) Base | Ours | Max@R-norm Base | Ours | Belebele Comp@10 Base | Ours | Max@R (↓) Base | Ours | Max@R-norm Base | Ours |
|---|---|---|---|---|---|---|---|---|---|---|---|---|---|
| *multilingual-e5-base* | | | | | | | | | | | | | |
| En+De | En | 54.20 | **65.04** | 24.16 | **16.10** | 64.82 | **70.55** | 66.89 | **88.33** | 52.44 | **13.98** | 51.98 | **71.41** |
| | De | 39.41 | **62.35** | 45.38 | **21.00** | 55.91 | **66.80** | 53.00 | **86.44** | 70.03 | **16.05** | 47.73 | **69.38** |
| En+El | En | 24.45 | **62.10** | 63.15 | **17.59** | 51.25 | **69.30** | 33.67 | **86.56** | 140.91 | **15.18** | 37.45 | **70.20** |
| | El | 7.73 | **53.18** | 248.40 | **30.10** | 31.91 | **61.71** | 11.78 | **76.78** | 287.99 | **22.93** | 26.94 | **64.14** |
| En+Hi | En | 46.97 | **62.61** | 26.87 | **18.84** | 63.32 | **68.33** | 65.33 | **87.89** | 47.43 | **14.08** | 53.46 | **71.31** |
| | Hi | 26.55 | **53.70** | 101.12 | **30.74** | 44.60 | **61.41** | 45.78 | **76.00** | 86.64 | **27.37** | 44.60 | **61.54** |
| En+Ro | En | 53.61 | **65.45** | 23.14 | **15.57** | 65.43 | **71.03** | 64.22 | **89.11** | 55.41 | **14.91** | 51.17 | **70.47** |
| | Ro | 23.28 | **61.01** | 89.76 | **24.89** | 46.28 | **64.40** | 39.56 | **82.44** | 145.65 | **20.94** | 39.96 | **65.48** |
| En+Tu | En | 42.44 | **62.69** | 35.54 | **17.82** | 59.37 | **69.12** | 59.11 | **87.51** | 10.04 | **7.51** | 76.28 | **80.54** |
| | Tu | 24.20 | **55.21** | 94.54 | **30.75** | 45.55 | **61.41** | 44.22 | **80.00** | 126.08 | **24.93** | 39.08 | **62.91** |
| *gte-multilingual-base* | | | | | | | | | | | | | |
| En+De | En | 75.13 | **75.71** | 10.45 | **9.86** | 76.65 | **77.48** | 88.56 | **92.67** | 8.79 | **6.99** | 78.23 | **81.61** |
| | De | 67.90 | **71.09** | 16.27 | **14.08** | 70.40 | **72.45** | 87.67 | **89.22** | 13.30 | **9.79** | 72.15 | **76.66** |
| En+El | En | 67.14 | **70.84** | 11.95 | **10.98** | 74.75 | **75.95** | 88.67 | **90.22** | 10.34 | **8.44** | 75.85 | **78.84** |
| | El | 51.68 | **55.71** | 25.08 | **20.28** | 64.29 | **67.29** | 81.00 | **84.33** | 24.88 | **18.72** | 62.94 | **67.12** |
| En+Hi | En | 71.93 | **72.27** | 12.07 | **11.52** | 74.61 | **75.28** | 89.00 | **91.67** | 9.35 | **8.82** | 77.33 | **78.18** |
| | Hi | 54.96 | **58.66** | 27.76 | **23.51** | 62.86 | **65.20** | 82.11 | **84.00** | 23.81 | **19.66** | 63.59 | **66.40** |
| En+Ro | En | 73.70 | **76.47** | 11.04 | **9.60** | 75.88 | **77.85** | 89.22 | **92.89** | 9.28 | **7.19** | 77.44 | **81.20** |
| | Ro | 64.54 | **67.56** | 16.94 | **14.13** | 69.83 | **72.40** | 84.89 | **88.22** | 17.44 | **12.57** | 68.17 | **72.97** |
| En+Tu | En | 69.24 | **71.68** | 12.82 | **11.03** | 73.76 | **75.89** | 90.22 | **92.33** | 10.04 | **7.51** | 76.28 | **80.54** |
| | Tu | 55.29 | **61.51** | 26.18 | **20.31** | 63.68 | **67.27** | 83.22 | **86.89** | 19.65 | **14.53** | 66.41 | **70.85** |
| *jina-embeddings-v3* | | | | | | | | | | | | | |
| En+De | En | 69.50 | **75.80** | 15.12 | **10.08** | 71.44 | **77.16** | 89.00 | **92.89** | 14.25 | **9.00** | 71.14 | **77.88** |
| | De | 72.86 | **74.79** | 13.94 | **11.08** | 72.58 | **75.82** | 91.33 | **92.33** | 11.16 | **9.30** | 74.75 | **77.41** |
| En+El | En | 53.03 | **71.76** | 31.84 | **11.98** | 60.93 | **74.73** | 80.56 | **89.22** | 36.69 | **13.11** | 57.21 | **72.35** |
| | El | 57.31 | **67.31** | 27.76 | **16.49** | 62.86 | **70.21** | 83.22 | **87.33** | 32.88 | **22.58** | 58.83 | **64.37** |
| En+Hi | En | 61.93 | **74.54** | 20.27 | **10.88** | 67.30 | **76.00** | 87.56 | **91.67** | 20.73 | **11.15** | 65.62 | **74.74** |
| | Hi | 62.86 | **69.58** | 24.82 | **16.52** | 64.43 | **70.19** | 85.00 | **86.67** | 28.77 | **20.32** | 60.83 | **65.92** |
| En+Ro | En | 64.54 | **74.20** | 24.53 | **10.65** | 64.63 | **76.38** | 83.22 | **92.22** | 31.87 | **10.17** | 59.30 | **76.10** |
| | Ro | 65.55 | **70.76** | 19.67 | **12.90** | 67.73 | **73.68** | 86.78 | **91.00** | 25.46 | **14.12** | 62.59 | **71.27** |
| En+Tu | En | 62.44 | **73.87** | 21.33 | **10.54** | 66.58 | **76.53** | 86.67 | **93.00** | 18.45 | **9.39** | 67.35 | **77.27** |
| | Tu | 65.71 | **71.60** | 18.89 | **13.19** | 68.28 | **73.36** | 87.56 | **89.89** | 18.65 | **12.71** | 67.19 | **72.82** |
| *bge-m3* | | | | | | | | | | | | | |
| En+De | En | 76.13 | **77.48** | 10.02 | **9.96** | 77.24 | **77.33** | 93.00 | **93.78** | 9.40 | **8.54** | 77.25 | **78.66** |
| | De | **76.22** | 75.97 | **10.14** | 10.44 | **77.08** | 76.67 | 92.22 | **93.22** | 8.44 | **8.12** | 78.83 | **79.41** |
| En+El | En | 68.24 | **72.94** | 13.62 | **12.11** | 72.91 | **74.58** | 89.33 | **91.67** | 11.11 | **9.73** | 74.79 | **76.75** |
| | El | 68.32 | **70.84** | 13.23 | 13.23 | 73.32 | 73.32 | 86.22 | **87.89** | 17.75 | **14.01** | 67.91 | **71.38** |
| En+Hi | En | 69.66 | **72.86** | 11.96 | **11.55** | 74.75 | **75.24** | 89.89 | **91.44** | 11.10 | **10.04** | 74.80 | **76.29** |
| | Hi | 68.07 | **70.42** | 13.69 | **13.19** | 72.84 | **73.36** | 83.11 | **85.89** | 21.98 | **19.88** | 64.76 | **66.24** |
| En+Ro | En | 78.07 | **78.32** | 9.97 | **9.66** | 77.31 | **77.75** | 92.67 | **93.22** | 9.66 | **8.56** | 76.84 | **78.63** |
| | Ro | 74.03 | **75.13** | 11.66 | **11.17** | 75.10 | **75.71** | 91.22 | **92.67** | 10.13 | **9.41** | 76.15 | **77.23** |
| En+Tu | En | 73.28 | **74.79** | 11.48 | **11.25** | 75.32 | **75.61** | 91.56 | **92.11** | 9.52 | **8.46** | 77.07 | **78.79** |
| | Tu | 71.34 | **72.27** | 13.38 | **12.33** | 73.16 | **74.32** | 89.56 | **90.78** | 13.82 | **11.75** | 71.58 | **73.96** |

Table 5: Performance comparison under the **Mono scenario** for five additional languages using the jina-embeddings-v3 model: English / Target language queries.

| Lang | Model | XQuAD Mono-Same NDCG@1 | MRR | Mono-Cross NDCG@1 | MRR | Belebele Mono-Same NDCG@1 | MRR | Mono-Cross NDCG@1 | MRR |
|---|---|---|---|---|---|---|---|---|---|
| De | Base | 89.2/**90.0** | 0.933/**0.937** | 87.8/87.9 | 0.923/0.925 | **88.4/89.3** | 0.919/0.926 | 84.9/86.0 | 0.897/0.903 |
| | Ours | 90.6/88.4 | **0.943**/0.929 | **89.1/88.9** | **0.933/0.932** | 87.6/88.1 | 0.916/0.916 | **86.8/86.3** | **0.909/0.905** |
| El | Base | 89.2/80.5 | 0.933/0.873 | 79.5/81.7 | 0.863/0.884 | 88.4/83.6 | 0.919/0.883 | 80.1/78.3 | 0.852/0.840 |
| | Ours | **90.0/86.8** | **0.940/0.916** | **88.4/83.5** | **0.928/0.895** | **88.8/84.0** | **0.924/0.887** | **83.4/81.1** | **0.883/0.864** |
| Hi | Base | 89.2/83.3 | 0.933/0.891 | 81.6/84.4 | 0.883/0.899 | 88.4/81.1 | 0.919/0.860 | 82.3/79.4 | 0.875/0.852 |
| | Ours | **91.6/87.5** | **0.950/0.922** | **89.0/85.2** | **0.933/0.907** | **89.1/82.2** | **0.926/0.872** | **86.0/79.4** | **0.902/0.857** |
| Ro | Base | **89.2**/84.7 | **0.933**/0.899 | 82.4/**85.3** | 0.885/**0.909** | **88.4**/85.1 | **0.919**/0.892 | 81.7/82.6 | 0.869/0.875 |
| | Ours | 88.7/**86.5** | 0.932/**0.917** | **88.1**/85.2 | **0.927**/0.908 | 87.2/**87.0** | 0.914/**0.907** | **84.8/85.2** | **0.895/0.898** |
| Tr | Base | 89.2/84.9 | 0.933/0.901 | 82.2/84.8 | 0.883/0.903 | 88.4/85.9 | 0.919/0.896 | 82.0/82.2 | 0.874/0.874 |
| | Ours | **89.5/88.6** | **0.938/0.928** | **89.1/85.2** | **0.935/0.907** | **88.4/86.7** | **0.921/0.908** | **87.0/84.9** | **0.911/0.892** |

