# OpenReview forum: "Improving Semantic Proximity in Information Retrieval through Cross-Lingual Alignment"
_ICLR.cc/2026/Conference — ICLR 2026 Poster_

### Official Review · Reviewer_Mk7b · 2025-10-27

**Soundness:** 3
**Presentation:** 3
**Contribution:** 3
**Rating:** 6
**Confidence:** 4

**Summary:**

This paper tackles English‑centric bias and poor cross‑lingual alignment in multilingual retrieval models. When English and non‑English documents coexist, models often favor irrelevant English content. To address this, the authors propose a mixed‑language evaluation setup, a new metric Max@R to measure alignment quality, and a joint training strategy combining Jensen–Shannon Divergence for embedding alignment with InfoNCE for contrastive retrieval learning. Experiments on XQuAD and Belebele show significant performance gains.

**Strengths:**

- The authors identify a critical limitation in existing CLIR models: when retrieving from multilingual document pools, multilingual embedding models tend to prioritize irrelevant English documents while overlooking relevant ones written in the same language as the query.
- The authors propose a new evaluation metric, Max@R, to measure retrieval performance more effectively in mixed-language document environments.
- The authors present an optimization approach that jointly uses embedding alignment and contrastive learning, which is straightforward and effective.
- Experimental results demonstrate the effectiveness of the proposed approach.

**Weaknesses:**

- The construction of alignment samples relies on translations generated by large language models, which may introduce translation bias or noise into the training data.
- The constructed dataset contains only 2,800 samples, which may lead to potential overfitting of the model.
- The proposed method is primarily trained with English queries. It is recommended to further explore retrieval performance using queries in a broader range of languages.

**Questions:**

See the Weaknesses.

---

> ### Author Response · Authors · 2025-11-16
> **Response to Reviewer Mk7b**
>
> We sincerely thank the reviewer for their insightful feedback and for recognizing the core contributions of our work, namely the Max@R metric and our joint training strategy. We appreciate the opportunity to address the valid points raised regarding potential weaknesses.
>
>  $\newline$
>
>
> ## Response to Weakness1
>
> We acknowledge the possibility of translation bias or noise being introduced while constructing the training data using an LLM (GPT-4o). We recognize this as a limitation of our study and have transparently disclosed it in the Limitations and Ethics Statement sections. While human translation is ideal, as the reviewer suggested, building thousands of parallel data samples across 10 languages incurs prohibitive costs and time.
> What we wish to emphasize is that our evaluation benchmarks, XQuAD and Belebele, are high-quality, human-translated datasets generated by professional translators (Section 5.1 , Appendix B). The fact that our model, despite being trained on (potentially noisy) machine-translated data, consistently improved performance over baseline models on these rigorous, human-translated benchmarks validates that our approach achieves meaningful and generalizable performance improvements under practical constraints.
>
>  $\newline$
>
> ## Response to Weakness 2
> As we demonstrated in Figure 1, existing multilingual embedding models possess a persistent, inherent "English bias". Our key finding is that our method can significantly ameliorate this bias and effectively realign the embedding space using only 2.8k samples. We argue this highlights the data efficiency and effectiveness of our methodology.
> If the model had merely overfit to these 2.8k samples, we would expect to see performance degradation in monolingual scenarios. However, our results demonstrate the opposite. As shown in the Mono scenario results in Table 2 and Table 5 , our model maintains or even slightly improves upon the baseline performance in both Mono-Same and Mono-Cross settings . This evidence suggests that our method did not simply overfit to the training data, but rather contributed to improving the generalized cross-lingual alignment capability of the embedding space.
>
>  $\newline$
>
> ## Response to Weakness 3
> Our proposed method is indeed trained with English as a central pivot. This was an intentional design choice to address the most critical and frequently encountered scenario in real-world multilingual environments: cross-lingual retrieval pivoted around English . Our core strategy was to directly align the embedding distributions of English documents ($p_{en}$) and target-language documents ($p_{tgt}$) via $L_{JSD}$, and to enhance the retrieval performance between English queries ($q_{en}$) and target-language documents ($p_{tgt}$) via $L_{NCE}$ .
> Our evaluation is structured around combinations of English and various target languages (e.g., En+Ar, En+Zh). Crucially, as Tables 1 and 4, and Figure 1(b) clearly demonstrate, our method achieved consistent and substantial performance gains over the baseline not only for English queries (which were used in training) but also for all target-language queries (e.g., "Query: Ar", "Query: Zh"). This suggests that our alignment mechanism did not merely learn a unidirectional relationship (English $\rightarrow$ target language), but rather improved the fundamental alignment of the embedding space itself, which generalized bidirectionally.
>
> Furthermore, the cross-lingual alignment mechanism proposed in this paper is designed to be language-agnostic. Therefore, we expect it to yield performance improvements not only in English-target language pairs but also in other cross-lingual combinations (e.g., non-English $\leftrightarrow$ non-English).
>
> $\newline$
>
> We thank you once again for your constructive and insightful review. We hope our explanations have successfully addressed the concerns you raised.

---

### Official Review · Reviewer_gRdR · 2025-10-27

**Soundness:** 2
**Presentation:** 2
**Contribution:** 2
**Rating:** 4
**Confidence:** 3

**Summary:**

The paper addresses challenges in Cross-Lingual Information Retrieval (CLIR) and Multilingual Information Retrieval (MLIR) tasks where multilingual embeddings often exhibit biases, especially toward English.The authors introduce a new evaluation metric (Max@R) and propose a training strategy combining Jensen-Shannon Divergence (JSD) and InfoNCE contrastive loss to improve the cross-lingual alignment of embeddings. The method demonstrates significant improvements in retrieval performance even with a small dataset of 2.8K samples. Experiments on XQuAD and Belebele datasets across 10 languages and 4 models demonstrate consistent improvements and reduced language bias disparities.

**Strengths:**

- 1. Proposing a cross-lingual embedding optimization method that combines JSD and InfoNCE, which is a novel and reasonable approach;

- 2. The experiment covers multiple metric and text embedding models.The experimental results are sufficient and the analysis of the experimental results is also very convincing.

- 3. Using only 2.8k samples to improve multiple pre-trained models is efficient and suggests the method is data-efficient for practitioners with limited parallel resources.

**Weaknesses:**

- 1. In the methodology section, especially the description of the JSD semantic alignment process is too brief, and readers may need diagrams to understand the idea of distribution level alignment. Equation 6 is motivated only by satisfying distance axioms, but many divergence measures satisfy these properties. The paper does not explain why JSD over, say, Wasserstein distance, Bhattacharyya divergence, or direct embedding-space alignment (e.g., via cosine similarity matching or optimal transport on embeddings) would be superior.

- 2. The lack of a more detailed description of the dataset has affected the intuitiveness of the experimental results. XQuAD and Belebele are QA datasets converted to retrieval by treating passages as documents. This is somewhat artificial; natural retrieval corpora with mixed languages would be more convincing.

- 3. More detailed explanations are needed for some metric, such as numerical ranges, numerical size meanings, etc.

**Questions:**

- Why is JSD in Equation 6 superior to alternative divergence measures for the retrieval task?

- In Table 1, the Comp@10 values range from approximately 0.5 to 93.22 (e.g., multilingual-e5-base En+Zh: 0.50 vs. gte-multilingual-base En+Ar: 87.44). Are these values reported as percentages (0–100 scale), or as proportions (0–1 scale)?

- Additionally, was the result averaged across multiple runs? If so, how many independent experiments were performed, and how were random seeds selected and fixed? Please provide clearer or additional explanations to improve reproducibility.

---

> ### Author Response · Authors · 2025-11-16
> **Response to Reviewer gRdR (1)**
>
> We sincerely thank the reviewer for the insightful and constructive feedback.
> Your comments have been very helpful in refining our explanations and strengthening the clarity and rigor of the paper.
>
>
> $\newline$
>
>
> ## Response to Weakness1 & Question :
> ### Visualization and Clarification of the JSD Alignment Process
>
> The figure we add an illustration that intuitively demonstrates the alignment effect within the embedding space achieved through JSD-based alignment, showing the results of steps (a)-(c):
>
> (a) An English-target language document pair ($p_{en}, p_{tgt}$) is mapped to vectors ($z_{d}^{en}, z_{d}^{tgt}$) in the shared embedding space.
>
> (b) Each embedding vector is interpreted as a probability distribution ($P(z_{d}^{en}), P(z_{d}^{tgt})$) via a softmax function .
>
> (c) Finally, the JSD-based distance ($\sqrt{\mathrm{JSD}(P \,\|\, Q)}$) between the two distributions is minimized, aligning them across all embedding dimensions between the languages .
>
> Furthermore, we will enhance the explanation in the main text to clarify that while the existing $L_{NCE}$ focuses on 'point-wise similarity' between individual samples, our proposed $L_{JSD}$ treats embeddings as probability distributions and performs a 'distribution-level alignment' role .
>
>
> $\newline$
>
> ---
>
> $\newline$
>
> ### Justification for Choosing JSD: Comparison with Other Divergence Measures
> We chose JSD in Equation (6) because we judged it to be the most practical choice for our high-dimensional, multilingual retrieval environment, considering computational efficiency, numerical stability, a global bound, and complementarity with the existing InfoNCE loss simultaneously. Compared to the Wasserstein distance or Bhattacharyya divergence mentioned by the reviewer, JSD is computationally lighter, its value scale is easier to interpret, and it exhibits more stable characteristics from a training perspective.
>
> (a) **vs. Wasserstein Distance:** Wasserstein distance requires solving an optimal transport (OT) problem, which is computationally very expensive ($O(n^2)$ or more) in our high-dimensional embedding ($dim=768$)  and large-batch environment. It also carries the burden of designing an additional 'ground cost' $c(i,j)$ between dimensions. Furthermore, unlike JSD, the Wasserstein value is not set by a simple global upper bound, so its practical value range can fluctuate significantly depending on the embedding scale and data distribution. This characteristic can make it difficult to set a stable learning rate and schedule, as the large distance values from certain query-document pairs can excessively dominate the gradients.
>
> $\newline$
>
> (b) **vs. Bhattacharyya Divergence:** This measure, defined as $D_B(P, Q) = -\ln \left( \sum_i \sqrt{P_i Q_i} \right)$, can diverge to infinity as the overlap between the two distributions ($\sum \sqrt{PQ}$) approaches zero (i.e., for very dissimilar documents). When normalizing high-dimensional embeddings with softmax, many dimensions will have very small probabilities, leading to a high risk of numerical instability and skewed losses for pairs with low overlap.
>
> $\newline$
>
> (c) **Advantages of JSD:** In contrast, JSD computes KL divergence via a mixture $M = \tfrac12 (P + Q)$ (i.e., $\mathrm{JSD}(P \,\|\, Q) = \tfrac12 \mathrm{KL}(P \,\|\, M) + \tfrac12 \mathrm{KL}(Q \,\|\, M)$). This mixture distribution $M$ always ensures $M_i > 0$, making it numerically stable. Moreover, its value is globally bounded in $[0, \log 2]$, meaning the scale remains consistent even for different query-document pairs. This is a key advantage for stably managing the loss and gradients during training.
>
> $\newline$
>
> (d) **vs. Direct Alignment (Cosine, etc.)**: The 'point-wise' alignment, such as cosine similarity matching mentioned by the reviewer, is the role already performed by the $L_{NCE}$ in our model . $L_{JSD}$ does not replace this; rather, it acts as a 'complementary distribution-level regularizer' that aligns the semantic roles of the embedding dimensions themselves across languages while $L_{NCE}$ aligns the individual samples. In our Ablation Study (Table 3), removing $L_{JSD}$ consistently degrades retrieval performance, which supports that our proposed distribution-level alignment is not just a design choice but a critical component contributing to the actual improvement in retrieval quality.
>
>
> $\newline$
>
>
> Reflecting these points, we (1) add the figure and (2) clearly describe the justification for using JSD.

---

> ### Author Response · Authors · 2025-11-16
> **Response to Reviewer gRdR (2)**
>
> ## Response to Weakness 2: Dataset description and artificiality concerns.
>
> We selected these two datasets specifically because a "fully parallel" structure was essential to rigorously validate our proposed 'cross-lingual alignment' and the new 'Max@R' metric . Our 'Multi' scenario assumes the existence of two ground-truth documents (one in English, one in the target language) for every query, and XQuAD and Belebele are high-quality benchmarks that allow us to create this specific experimental environment.
>
> We understand why this setup might appear "artificial." However, we would like to highlight that this is a standard and widely accepted approach in the multilingual embedding evaluation field. In fact, the MMTEB[1] , a key benchmark in the field, also uses XQuAD and Belebele as core benchmarks for multilingual retrieval evaluation, which can be verified in the official MTEB library[2].
>
> Regarding the concern about a lack of detailed description, we have provided a comprehensive explanation of these benchmarks in Appendix B. EVALUATION BENCHMARK DETAILS . This section specifies that XQuAD and Belebele are not based on simple machine translation, but are high-quality, human-translated datasets generated by professional translators.
>
> We take the reviewer's feedback to improve clarity. In the revised version, we add a reference to Appendix B in the main body (Section 5.1), and we also enhance Appendix B to provide a more detailed description of the datasets. This will make our rationale for the experimental design clearer to the reader.
>
>
> $\newline$
> ### References
> [1] MMTEB: Massive Multilingual Text Embedding Benchmark. ICLR 2025
>
> [2] https://github.com/embeddings-benchmark/mteb
>
>
> $\newline$
>
>
> ## Response to Weakness 3 & Question 2: Clarification of the Comp@10 metric.
>
> The Comp@K values are indeed reported as percentages on a 0-100 scale, as the reviewer correctly surmised. Therefore, the 0.50 score recorded by the multilingual-e5-base model for En+Zh (when using a target language query) signifies that only 0.50% of queries successfully retrieved both parallel ground-truth documents within the Top-10 results, indicating extremely poor performance. This figure is a stark example of the severe 'English bias' and 'cross-lingual alignment failure' in existing models, which we pointed out in Section 3.1. To clarify this point and prevent future confusion, we amend the 'Metric' of Section 5.1 in the revised version. We explicitly state that Comp@10 is a percentage (0-100 scale) and describe its meaning in more detail.
>
>
> $\newline$
>
>
> ## Response to Question 3: Reproducibility and the use of random seeds.
> We acknowledge that stochastic elements during the training process can influence the final performance. All experimental results reported in this paper are based on a single run using a fixed, single random seed (42).
> This was an intentional design choice to ensure a fair and consistent comparison between all baselines and our proposed model. This fixed seed controls all stochasticity, including data shuffling, and the batch composition and sampling order for each language-specific experiment.
>
> In other words, we strictly controlled the experimental environment by using the exact same training settings and batch configurations when training the baseline models and our proposed model. This ensures that the comparison fairly isolates the effect of our methodology. We explicitly add these reproducibility details to Appendix C (Training Details), as suggested by the reviewer .
>
> $\newline$
>
> We sincerely appreciate your valuable feedback and would be happy to clarify further if any additional questions arise.

---

### Official Review · Reviewer_dUrW · 2025-10-31

**Soundness:** 2
**Presentation:** 2
**Contribution:** 2
**Rating:** 4
**Confidence:** 2

**Summary:**

This paper introduces a new training strategy to enhance cross-lingual alignment in information retrieval, addressing the issue of language bias and semantic misalignment in multilingual document pools. The authors propose a method that combines Jensen-Shannon Divergence (JSD) and InfoNCE losses to improve semantic alignment and retrieval performance. By aligning semantic embeddings across languages and optimizing retrieval capabilities, their approach significantly reduces language bias and improves retrieval accuracy. Experiments across multiple languages and datasets demonstrate the effectiveness of the proposed method, showing substantial improvements in cross-lingual retrieval performance and reduced bias, thus providing a robust solution for enhancing semantic proximity in multilingual information retrieval.

**Strengths:**

1. The writing of this work is relatively clear.
2. The experimental validation in this work is relatively comprehensive.

**Weaknesses:**

1. In the section “Introduction”, why does this paper deliberately emphasize English rather than other languages?

2. In the section “Related work”, ① The description "However, most prior studies assume either purely monolingual or entirely multilingual document pool, thus failing to adequately address biases and misalignments that arise when two languages coexist in a single document pool." seems ambiguous. The focus of this paper should also be on multilingual issues, and I am not clear about the differences between the current description and the methods mentioned. ② The descriptions "While insightful, these works primarily emphasize alignment in general representation tasks, offering limited consideration of practical retrieval challenges associated with combined-language environments" are vague and inaccurate, and the authors are advised to clarify them. ③ I also noticed that some works [1][2] seem similar to this paper, but they are not introduced or reviewed. What are the differences between [1] and this paper?

3. In the section “Method”, how is the positive sample pair defined?

**Reference**

[1]  Zuo Y, Jiang W, Liu W, et al. Alignxie: Improving multilingual information extraction by cross-lingual alignment[J]. arXiv e-prints, 2024: arXiv : 2411.04794.

[2]  Kargaran A H, Modarressi A, Nikeghbal N, et al. MEXA: Multilingual evaluation of English-centric LLMs via cross-lingual alignment[J]. arXiv preprint arXiv:2410.05873, 2024.

**Questions:**

Please refer to the “Weakness” section for related questions.

---

> ### Author Response · Authors · 2025-11-16
> **Response to Reviewer dUrW (1)**
>
> We sincerely thank the reviewers for their thoughtful reading of our paper and valuable feedback. Below, we provide point-by-point responses to each identified Weakness and related question. We hope our clarifications address the points raised.
>
> $\newline$
>
> ## Response to Weakness 1
> As the reviewer correctly observed, our work deliberately emphasizes "English." This is because, in real-world cross-lingual retrieval scenarios, configurations where English is combined with another target language are both the most critical and the most frequent in practice. As we point out in the Introduction, current multilingual models suffer from a severe "English inclination problem," where they tend to rank unrelated English documents above relevant documents written in the same language as the query. To directly address this realistic failure mode, we intentionally center our experimental setup around English, designing our method and evaluation specifically to diagnose and mitigate this English-centric bias.
>
>
> $\newline$
>
>
> ## Response to Weakness 2-1 & 2-2
>
> As we describe in Section 2 (Preliminary), our intention here is to highlight a fundamental difference between our new evaluation scenario and conventional cross-lingual retrieval settings. As the reviewer notes, both settings are multilingual in a broad sense, but they differ crucially in how the "ground-truth" is defined and evaluated.
>
>
> - **Conventional CLIR (Cross-Lingual Information Retrieval)**:
>   The query language ($L_q$) and the document collection language ($L_d$) are different ($L_q \ne L_d$). For example, an English (L1) query is used to retrieve documents from a collection that consists only of Chinese (L2) documents.
>
> - **Conventional MLIR (Multilingual Information Retrieval)**:
>   A query in language L1 is used to retrieve relevant documents from a collection that contains multiple languages (L2, L3, L4, …), and the goal is to find L1-relevant documents (possibly in multiple languages) in this mixed-language pool.
>
> - **Our scenario (multi-reference in CLIR)**:
>   Our core assumption is that, in the cross-lingual environment of interest, there **exist two parallel ground-truth documents** for each query: one in English and one in the target language (e.g., one English relevant document and one target-language relevant document).
>
> The key issue we point out is that conventional CLIR and MLIR setups often **fail to expose deeper problems such as inaccurate cross-lingual alignment and language bias**. For example, conventional setups can separately evaluate how well a model retrieves the L2 ground-truth for an L1 query (CLIR), or how well it retrieves the L3 ground-truth for an L1 query (MLIR). However, they are not well-suited to directly measure whether, for a single L1 query, the model prefers an **irrelevant L1 document** over a **relevant L2 document** when both coexist in the same pool.
>
>
> Our scenario is specifically designed to:  (i) clearly expose such language bias, which is hard to diagnose in traditional setups, and  (ii) evaluate the model’s **overall retrieval performance** in terms of bringing **both** parallel ground-truth documents (English and target language) into the top ranks simultaneously.
>
>
> When we describe prior cross-lingual alignment work as focusing on "general representation tasks," our point is that many of these works concentrate on generic representation alignment, such as alignment at the sentence level based on semantic similarity. However, they do not explicitly model or evaluate the practical retrieval scenario that we investigate. In our setting, two equivalent answers written in different languages are both present, and the model must retrieve them together with minimal language bias. In other words, success in general alignment does not necessarily translate to success in our practical retrieval environment, where both low language bias and strong performance in retrieving both ground-truth documents are essential.

---

> ### Author Response · Authors · 2025-11-16
> **Response to Reviewer dUrW (2)**
>
> ## Response 2-3: Comparison with the suggested related works
>
> The reviewer points out that the suggested works [1] and [2] also consider multiple languages and discuss cross-lingual alignment, and therefore may look similar to our work at first glance. However, the problem setups, target tasks, model types (generative LLMs), and core contributions of these works are all quite different from our focus.
>
> In particular, both [1] and [2] are built around **generative LLMs**, while our paper focuses on retrieval performance in cross-lingual scenarios. We believe this makes them only indirectly related rather than directly comparable baselines. For clarity, we summarize the key ideas of [1] and [2] and how they differ from our work:
>
> [1] Zuo et al., "AlignXIE: Improving Multilingual Information Extraction by Cross-Lingual Alignment"
>
> - Improve multilingual Information Extraction (IE) performance.
> - Unify IE tasks across multiple languages as a code generation problem, and use Python class–based schema representations to normalize and align heterogeneous IE schemas across languages.
> - They propose a translated instance prediction task and a parallel IE dataset (ParallelNER), and use a cross-lingual alignment stage specialized for IE to fine-tune LLM-based generative models, thereby reducing performance gaps across languages.
> - In short, AlignXIE focuses on multilingual IE with a **code-generation-style generative LLM**, rather than on dense retrieval ranking.
>
> [2] Kargaran et al., "MEXA: Multilingual Evaluation of English-Centric LLMs via Cross-Lingual Alignment"
>
> - Propose an evaluation methodology for assessing the multilingual capability of English-centric LLMs.
> - Using sentence-level parallel corpora such as FLORES-200 and the Bible, they compare the representations of English vs. non-English sentences and compute an alignment score for each language that reflects its proximity to English in the model’s representation space.
> - They show that this alignment score correlates strongly with performance on multilingual downstream benchmarks such as Belebele, m-MMLU, and m-ARC, thus validating MEXA as a metric for estimating LLM multilinguality.
> - The models considered are English-centric generative LLMs such as Llama, Gemma, Mistral, etc. Thus, [2] is an evaluation framework for **generative LLMs**, not for dense retrievers.
>
> In summary, [1] focuses on improving cross-lingual alignment for multilingual IE with generative LLMs, and [2] proposes a metric for evaluating the multilingual capabilities of English-centric generative LLMs. In contrast, our work targets **cross-lingual information retrieval with multilingual dense retrievers**, specifically addressing English bias and proposing retrieval-oriented metrics and training objectives for this scenario.
>
>
> $\newline$
>
>
>
> ## Response to Weakness 3
> The reviewer asks how positive sample pairs are defined in our method. This is described in Section 4 of the paper, especially in Sections 4.1 and 4.2, but we acknowledge that it may not have been sufficiently explicit and will clarify it in the revision.
>
> We use training data of the form $(q_{en}, p_{en}, p_{tgt})$, where $(q_{en})$ is an English query, $(p_{en})$ is the English ground-truth document, and $(p_{tgt})$ is the target-language ground-truth document that is semantically equivalent to $(p_{en})$. Based on this, we define two distinct types of positive pairs:
>
> - Positive pair for $L_{JSD}$ (alignment loss):
>   To directly align the embedding distributions across languages, we treat the **two semantically equivalent documents**  $(p_{en}, p_{tgt})$ as a positive pair.
>
> - Positive pair for $L_{NCE}$ (retrieval loss):
>   To improve retrieval performance, we treat the **English query and the target-language ground-truth document** $(q_{en}, p_{tgt})$ as a positive pair.
>
>
> $\newline$
>
>
> We hope that these responses address the reviewer’s concerns and help clarify the contributions and scope of our work.

---

### Official Review · Reviewer_oW6S · 2025-11-01

**Soundness:** 3
**Presentation:** 3
**Contribution:** 3
**Rating:** 6
**Confidence:** 4

**Summary:**

In the paper, the authors revisit the Cross Lingual Information Retrieval (CLIR) task through the lens of cross-lingual alignment evaluation. They argue that in a CLIR setup, the English unrelated documents, for a given query, are prioritized in the ranked list when compared with the related documents in the same language as the query. To alleviate this, they have proposed both new metrics and training setups, testing on standard benchmarks. The results show that the proposed setup achieves better semantic alignment in the CLIR task across many languages.

**Strengths:**

The problem is clearly described. The formal presentation of the problem premise in Section 2 makes it easy to comprehend.

* The demonstration through Performance Disparities, esp. for Chinese, is insightful.

* The authors show semantic Representation Instability among languages. They show that the performance is worse for Arabic and Thai when compared with Spanish.

* The authors also show high retrieval ranks for many languages.

These three key findings establish the motivation for a better setup.

Based on the premise, the authors propose semantic alignment techniques to ensure better alignment between the query and the relevant English document. This is proposed by the application of InfoNCE between the query and the relevant document. JSD is deployed to minimize semantic differences between embedding vectors.

The authors report experiment results on datasets: XQuAD and Belebele using metrics like Complete@K. In total, 10 languages have been considered.

The results show increased cross-lingual alignment, showing the efficacy of the proposed training setup.

In section 5.4, the superior performance over LNCE is reported.

Most of the standard embeddings, like multilingual-e5, gte-multimodal, bge, etc., have been considered.

**Weaknesses:**

It is not clear why the existing metrics, like MAP, can not be used for CLIR evaluation. The new metric Max@R, needed a clearer comparison with existing IR evaluation platforms in the CLIR setup. The authors need to establish better why a new metric was needed and how the existing metrics (even with customization for the premise of the paper) were inadequate.

The setup "We report the performance achieved when querying in each language within these environments." (line 159-160) need more details for reproducibility and because, importantly, the claims in this section hinge on the reliability of these results. I will be curious to also look into word-overlap-based and yet effective baselines like BM25 in the CLIR setup, empowered with translation resources like https://github.com/facebookresearch/fairseq/tree/nllb (No Language Left Behind: Scaling Human-Centered Machine Translation).

Though a little older, I would still suggest comparison with Sentence-BERT: Sentence Embeddings using Siamese BERT-Networks (sentence-transformers/paraphrase-multilingual-MiniLM-L12-v2 and other multilingual variants).

**Questions:**

The motivation for a new metric needs clearer motivation.

More baselines, as already suggested, will further fortify the claim of the paper.

---

> ### Author Response · Authors · 2025-11-16
> **Response to Reviewer oW6S (1)**
>
> Thank you very much for dedicating your valuable time to providing such an in-depth review of our paper. We are sincerely grateful that you recognized and clearly highlighted the strengths and contributions of our work. We would like to address the specific points you raised in the "Weaknesses" section of your review:
>
>
> $\newline$
>
>
> ## Response to Weakness 1: Clarification on the new evaluation metric (Max@R)
> Existing metrics like MAP (Mean Average Precision) work excellently for standard Information Retrieval and traditional CLIR evaluation. However, our paper points out the limitations of conventional CLIR and proposes a new 'Multi-reference' scenario, where two parallel ground-truth documents (e.g., one in English and one in the target language) coexist for a single query.
> This scenario is designed to measure the 'query language bias' that existing metrics like MAP, MRR, or NDCG (which uses a fixed $k$) fail to capture. Our proposed metric, Max@R, measures the 'worst rank' required to find all defined ground-truth documents ($R_q$) —that is, the 'actual cutoff point' one must reach to retrieve both documents with 100% recall. Therefore, when Max@R returns a value like 650.95, it intuitively exposes the 'retrieval cost' and the 'degree of bias,' indicating that, on average, one must search through 650 documents to find the second ground-truth answer. In conclusion, Max@R is not intended to replace MAP. Instead, it serves as an essential diagnostic metric specifically for the 'Multi-reference' scenario defined in this paper, designed to diagnose the severe language bias that existing metrics obscure. We will add this clarification to Section 3.
>
>
> $\newline$
>
>
> ## Response to Weakness 2: Clarification on the Experimental Setup for Reproducibility
> Regarding the experimental setup mentioned in lines 159-160, we have revised the relevant paragraph(s) to ensure reproducibility.
>
>
> $\newline$
>
>
> ## Response to Weakness 3: Regarding the suggestion of a BM25 baseline
> The problem our paper addresses is the "cross-lingual semantic misalignment" and "English bias" of SOTA dense multilingual embeddings. This is an issue of semantic distribution mismatch within the embedding space, not a problem of superficial lexical matching (which BM25 addresses). Our goal is to 'repair' this internal flaw in semantic models using $L_{JSD}$.
> Conversely, "BM25 + NLLB translation" is a lexical-based approach that fundamentally bypasses the very problem we are trying to solve: the alignment of the multilingual embedding space itself. Furthermore, in our proposed "Multi-reference" scenario, the "BM25 + translation" method reveals practical limitations in terms of computation and cost.
> - Because BM25 is based on lexical matching, if a target language query ($q_{tgt}$) is used to search this single, mixed-language pool, $q_{tgt}$ will only find lexical matches with the target language document ($Doc_{Tgt}$). Its lexical overlap with the English document ($Doc_{En}$) will be zero.
> - Consequently, a single $q_{tgt}$ query can only find $Doc_{Tgt}$.
> - To find the second ground-truth document ($Doc_{En}$), one must translate the query ($q_{tgt} \rightarrow q_{en}$ using NLLB) and then perform a second search on the same mixed pool.
>
>
> This is a fundamentally different and inefficient paradigm compared to our dense model, which aims to retrieve both ground-truth documents ($E(Doc_{Tgt})$ and $E(Doc_{En})$) simultaneously in a single search pass, using only a single query embedding ($E(q_{tgt})$) without any translation.
>
>
> In conclusion, we believe that BM25+translation is not an appropriate baseline for validating our core hypothesis.

---

> ### Author Response · Authors · 2025-11-16
> **Response to Reviewer oW6S (2)**
>
> ## Response to Weakness 4: Experiment on Additional MutlilinBaseline (Sentence-BERT)
>
> We thank the reviewer for this constructive suggestion. While we initially felt our experiments, centered on SOTA multilingual embedding models, were sufficient, we have actively incorporated the reviewer's suggestion by conducting an additional experiment applying our same methodology to paraphrase-multilingual-MiniLM-L12-v2, a multilingual variant of SBERT. The tables below compare the baseline performance with the performance after applying our alignment methodology (Ours) in the 'Multi' scenario on the XQuAD and Belebele datasets.
>
>
>
> $\newline$
>
>
>
> ## XQUAD
>
> | Query Lang | c@10-base | c@10-ours | max@R-base | max@R-ours | max@R norm-base | max@R norm-ours |
> | ---------- | ---------------- | ---------------- | ------------------- | ------------------- | ------------------------ | ------------------------ |
> | **EN**     | 51.01            | **53.11**        | 39.69               | **36.76**           | 57.81                    | **58.89**                |
> | **AR**     | 39.16            | **41.85**        | 78.78               | **71.43**           | 48.13                    | **49.51**                |
> | **EN**     | 56.05            | **57.56**        | 32.77               | **30.62**           | 60.51                    | **61.47**                |
> | **ZH**     | 48.15            | **51.85**        | 43.42               | **38.51**           | 56.54                    | **58.23**                |
> | **EN**     | 60.84            | **61.18**        | 29.58               | **28.16**           | 61.96                    | **62.65**                |
> | **DE**     | 55.13            | **57.56**        | 39.02               | **36.47**           | 58.05                    | **59.00**                |
> | **EN**     | 53.45            | **53.87**        | 39.83               | **37.65**           | 57.76                    | **58.55**                |
> | **EL**     | 46.81            | **49.08**        | 54.79               | **49.05**           | 53.25                    | **54.82**                |
> | **EN**     | 53.28            | **53.28**        | 38.72               | **38.07**           | 58.16                    | **58.40**                |
> | **HI**     | 45.71            | **47.65**        | 70.71               | **62.99**           | 49.65                    | **51.29**                |
> | **EN**     | **60.34**        | 59.83            | 31.05               | **29.53**           | 61.27                    | **61.98**                |
> | **RO**     | 54.12            | **55.71**        | 41.03               | **38.03**           | 57.34                    | **58.41**                |
> | **EN**     | 60.25            | **62.35**        | 29.57               | **27.31**           | 61.96                    | **63.09**                |
> | **ES**     | 56.89            | **58.99**        | 42.46               | **37.40**           | 56.85                    | **58.65**                |
> | **EN**     | 53.36            | **53.95**        | 36.65               | **34.08**           | 58.93                    | **59.96**                |
> | **TH**     | 43.03            | **47.39**        | 56.54               | **49.13**           | 52.81                    | **54.79**                |
> | **EN**     | 57.82            | **58.32**        | 32.14               | **29.68**           | 60.79                    | **61.91**                |
> | **TU**     | 51.43            | **52.27**        | 44.46               | **43.09**           | 56.20                    | **56.65**                |
> | **EN**     | **58.49**        | 57.48            | 32.64               | **31.01**           | 60.57                    | **61.29**                |
> | **VI**     | 52.94            | **53.03**        | 44.96               | **41.90**           | 56.05                    | **57.04**
>
> ---

---

> ### Author Response · Authors · 2025-11-16
> **Response to Reviewer oW6S (3)**
>
> ## BELEBELE
>
> | Query Lang | c@10-base | c@10-ours | max@R-base | max@R-ours | max@R norm-base | max@R norm-ours |
> | ---------- | ---------------- | ---------------- | ------------------- | ------------------- | ------------------------ | ------------------------ |
> | **EN**     | 73.78            | **74.44**        | 36.66               | **33.68**           | 57.24                    | **58.49**                |
> | **AR**     | 61.11            | **64.33**        | 69.95               | **59.95**           | 47.75                    | **50.01**                |
> | **EN**     | 77.67            | **79.44**        | 27.37               | **26.54**           | 61.54                    | **61.99**                |
> | **ZH**     | 73.44            | **77.33**        | 33.60               | **31.02**           | 58.52                    | **59.70**                |
> | **EN**     | **81.00**        | 80.78            | 23.75               | **23.29**           | 63.62                    | **63.91**                |
> | **DE**     | 77.22            | **78.44**        | 30.17               | **27.87**           | 60.11                    | **61.27**                |
> | **EN**     | 77.00            | **77.22**        | 34.78           | **34.09**               | 58.02                | **58.31**                    |
> | **EL**     | 69.78            | **72.22**        | 46.68               | **43.83**           | 53.69                    | **54.62**                |
> | **EN**     | 74.56            | **75.44**        | 32.69               | **31.04**           | 58.93                    | **59.69**                |
> | **HI**     | 60.22            | **62.00**        | 69.12               | **61.47**           | 47.92                    | **49.64**                |
> | **EN**     | 80.00            | **80.44**        | 25.42               | **24.74**           | 62.63                    | **63.02**                |
> | **RO**     | 75.56            | **76.56**        | 35.54               | **33.26**           | 57.70                    | **58.67**                |
> | **EN**     | 78.89            | **80.44**        | 30.64               | **27.54**           | 59.88                    | **61.45**                |
> | **ES**     | 74.00            | **77.11**        | 44.78               | **37.37**           | 54.30                    | **56.96**                |
> | **EN**     | **76.44**        | 76.33            | 29.56               | **27.50**           | 60.41                    | **61.47**                |
> | **TH**     | 65.00            | **67.56**        | 62.25               | **54.19**           | 49.46                    | **51.50**                |
> | **EN**     | 79.11            | **80.22**        | 23.88               | **23.27**           | 63.54                    | **63.92**                |
> | **TU**     | 73.11            | **73.67**        | 42.59               | **40.21**           | 55.04                    | **55.88**                |
> | **EN**     | **79.11**        | 78.67            | 23.15           | **23.05**               | 64.00                    | **64.07**                |
> | **VI**     | 75.44            | **76.78**        | 30.82               | **28.09**           | 59.79                    | **61.16**                |
>
> ---
>
>
> $\newline$
>
>
> As the tables clearly show, this model also exhibits the 'semantic misalignment' and 'English bias' issues we identified in Section 3.1 (e.g., on Belebele, the AR query Max@R is 69.95 vs. the En query's 36.66).
> However, when we applied our proposed $L_{JSD} + L_{NCE}$ alignment strategy (Ours), the model's performance was also consistently improved, as demonstrated by the results (e.g., Belebele AR query Max@R $69.95 \rightarrow \mathbf{59.95}$, c@10 $61.11\% \rightarrow \mathbf{64.33\%}$). Through this, we could confirm that the model's performance was consistently improved simultaneously with alignment.
>
>
> $\newline$
>
>
> Thank you very much for your thoughtful review and helpful feedback. We would be happy to clarify or discuss further if you have any additional questions.

---

### Author Response · Authors · 2025-11-17
**Global Response**

We sincerely thank the reviewers for their detailed and constructive feedback. In this round of response, we have made the following revisions, with newly added or modified parts indicated in blue font.

$\newline$

### (1) Clarification of Max@R and the Multi-reference setting
>  We have added and refined an explanation in Section 3 that Max@R is a diagnostic metric that measures "the actual cutoff rank required to retrieve both parallel ground-truth answers." We also clarify that, unlike conventional metrics, Max@R directly reveals language bias and the rank gap between ground-truth answers.

$\newline$

### (2) Experimental setup, metrics, and reproducibility
> We explicitly describe the configuration of the Multi / Multi-1 / Mono scenarios, the query language settings, and clarify that Comp@10 is a ratio-based metric on a 0–100 scale. We also add concrete details related to reproducibility—such as the single random seed (42) and training hyperparameters—in Appendix C.

$\newline$

### (3) Enhanced dataset description
> We more clearly describe the characteristics of XQuAD and Belebele as fully parallel datasets and provide citations in the main text and Appendix B. We emphasize that these benchmarks allow us to construct stable parallel ground-truth pairs across diverse language pairs, thereby supporting the validity of our proposed Multi-reference setting and multilingual experiments.

$\newline$

### (4) Further explanation of methodology and limitations
> We newly visualize the outcome of our training method, including the JSD-based distribution alignment process, in Figure 1, and provide additional explanation that simply maximizing similarity scores in standard CLIR objectives is insufficient to guarantee robust semantic proximity. We revise the main text to emphasize that even when a query–document pair attains the same similarity score, there can still be misalignment at the distribution level, and that aligning the embedding distributions themselves is necessary to resolve this issue.


$\newline$

We also addressed other minor issues. We will detail them in the individual response.


We believe our revision should address the concerns. However, please let us know any further suggestions and concerns, and we are willing to revise accordingly!

---

### Author Response · Authors · 2025-12-03
**Sincere Gratitude to Our Reviewers**

We sincerely regret that the continuity of the in-depth rebuttal and revision process we were conducting with the reviewers has been disrupted due to the recent system issues. Although we regret that we can no longer receive additional feedback from the reviewers, we would like to express our Great Thanks once again to the reviewers for recognizing the value of our work and providing constructive feedback. We have faithfully incorporated all concerns raised during the discussion period into the revision.



Sincerely,



The Authors

---

### Meta-Review · Area_Chair_wx1t · 2026-01-01

**Summary:**

Paper Summary. This paper identifies a limitation in cross-lingual alignment for information retrieval. They found that when the document pool contains documents in both English and other languages, the multilingual retrievers have a bias to select unrelated English documents over related documents in the same language as query. To address this, they propose a metric to quantify this cross-lingual alignment performance, and then design a training algorithm to enhance cross-lingual alignment.

Paper Strengths. The paper is well-motivated with a clear problem statement based on empirical evidence. The query language bias in CLIR is an insightful observation. The proposed approach combining JSD and contrastive learning is sound and data-efficient. They present solid and comprehensive experiments including 10 languages and several popular multilingual embeddings with substantial performance improvements.

Reviewer Concerns.
(1) More explanation of the new multi-reference evaluation setup compared to traditional CLIR and other related work should be provided. The necessity of a new metric, Max@R, is not established compared to existing metrics. (Reviewer oW6S, Reviewer dUrW).

(2) More baselines can be included BM25 and sentence-BERT (Reviewer oW6S)

(3) More details on experimental setting, dataset, and metric shall be provided for reproducibility (Reviewer oW6S, Reviewer gRdR)

(4) The emphasize on English in CLIR configure should be explained (Reviewer dUrW, Reviewer Mk7b)

(5) More details on methodology such as JSD semantic alignment and training data construction should be provided (Reviewer dUrW, Reviewer gRdR)

(6) The samples for training are constructed by LLMs which have translation noise and only contain limited samples. (Reviewer Mk7b).

(7) The proposed method is primarily trained with English queries, and it should be clarified if it improves queries in other languages (Reviewer Mk7b).

**Reviewer Concerns:**

Addressed Concerns.

(1) The author rebuttal justifies the motivation of the new metric Max@R for cross-lingual information retrieval settings with two ground-truth documents in two languages. It is designed to measure the query language bias, and existing metrics cannot capture this.

(2, partially) The author rebuttal added a new experiment on sentence-BERT, which demonstrated the effectiveness of their approach.

(3) (4) (5) (7) The author rebuttal also addressed several clarity issues regarding the new CLIR setting definition, difference from related work, training data construction, experimental settings.

Outstanding Concerns.

(2) The author rebuttal claims that BM25+translation is not an appropriate baseline, which I do not fully agree with. Both lexical and dense retrieval are appropriate and comparable solutions to CLIR, and the experiments should also discuss the contribution in the context of lexical-based retrieval. This is a weakness, yet the contribution of the paper still stands by addressing the limitation of multilingual embeddings.

(6) The author acknowledges the limitations of the translation bias in LLMs. This is a very minor concern to me.

**Reviewer Scores:**

Reviewer oW6S: 6 -> 6 (due to the outstanding concerns on BM25 baseline)

Reviewer dUrW: 4 -> 6

Reviewer gRdR: 4 -> 6

Reviewer Mk7b: 6 -> 6 or 8 (due to the limitation of the translation bias in LLMs)

---

### Decision · Program_Chairs · 2026-01-26

Accept (Poster)